# DIVERSE TEXT GENERATION THROUGH SOFT PROMPT TUNING

## ABSTRACT

Diverse text generation is crucial for effective exploration in language models. Current sampling-based decoding methods struggle to balance quality and diversity and lack control over generating mutually distinct outputs. Reinforcement learning approaches maintain quality, but require extensive training and are difficult to transfer across domains due to task-specific reward functions. We propose a lightweight framework that learns diversely initialized continuous soft prompt vectors, which, when prepended to input prompts, guide the model's final-token hidden states into distinct representation regions. This enables diverse generations from identical inputs, as initial hidden state differences amplify through the autoregressive mechanism, creating increasingly divergent generations. By preserving earlier hidden state similarities, our method maintains contextual consistency to task-specific constraints. Experiments across combinatorial tasks, question generation, and molecular design reveal that our soft prompt tuning method improves diversity while consistently adhering to task-specific constraints. Our approach shows particular strength in complex settings with large exploration spaces, as demonstrated through our novel contribution of a challenging combinatorial dataset specifically designed to evaluate diverse generation capabilities of language models. This lightweight framework provides a unified, broadly applicable solution for diverse text generation across various application domains.

## 1 INTRODUCTION

Recent advances have leveraged the power of language models (LLMs) for high-quality text generation tasks (Li et al., 2021; Becker et al., 2024). Beyond quality, many applications require diverse generation capabilities, including style transfer, open-ended storytelling, and creative content production (Jhamtani et al., 2017; Rao & Tetreault, 2018). Diverse generations not only provide an effective mechanism for data augmentation but, more importantly, unlock the potential of language models to perform meaningful exploration. This capability is desired for many applications, such as molecule generation for drug discovery and agent planning (Jang et al., 2025; Guan et al., 2023; Valmeekam et al., 2023; Singh et al., 2022).

A widely adopted strategy for promoting output diversity is to apply stochastic decoding techniques such as minimum-probability sampling (Minh et al., 2024), temperature scaling (Ackley et al., 1985), top-$k$ sampling (Fan et al., 2018), and nucleus (top-$p$) sampling (Holtzman et al., 2019). Despite their broad applicability, these methods suffer from notable shortcomings in both fidelity and distinctness. As diversity is increased, fluency and adherence to implicit task constraints often deteriorate (Shi et al., 2018; Du et al., 2022). Moreover, these approaches provide coarse control: when generating multiple responses, there is no explicit mechanism ensuring that the outputs are meaningfully distinct from one another, as diversity primarily stems from the randomness in sampling.

To encourage both quality and diversity, Reinforcement Learning (RL)–based methods have been explored as an alternative (Gou et al., 2023; Jang et al., 2025). These approaches can effectively promote exploration, but they are inherently domain-specific, requiring the careful design of task-dependent reward functions. This not only hinders cross-task generalization but also introduces substantial training overhead due to specialized optimization objectives. While RL offers greater controllability, it inevitably biases the generations toward characteristics encoded in the reward, thereby underrepresenting other aspects of diversity—particularly those not explicitly captured by

Figure 1: An example combinatorial task that requires diverse generation. Our method enables the LLM to generate diverse yet valid combinations for a given constraint.

the reward signal. As a result, the space of possible generations remains constrained, despite the increased optimization complexity.

To combine the best of both worlds, we introduce a lightweight and context-agnostic soft prompt tuning framework that steers generation directly toward diversity while preserving contextual consistency, without relying purely on randomness. Soft prompts are continuous learnable vectors that operate in the embedding space, guiding language model behavior without modifying the underlying weights (Lester et al., 2021). Our Soft Prompt Diversification approach optimizes multiple diversely initialized prompts, generated via scrambled Sobol sequences (Chi et al., 2005), to yield varied yet coherent outputs. During optimization, the model contrasts generations with and without soft prompts: it maximizes differences in final-token hidden states (promoting diversity) while minimizing deviations in earlier states (preserving context). Distinctions among final-token states across different prompts are further amplified to ensure mutually distinct generations. The expressive capacity of soft prompts has been demonstrated empirically, with a single vector $\mathbf{p} \in \mathbb{R}^d$ able to reconstruct text sequences of up to 1,000 tokens (Liu et al., 2025). Leveraging this representational power, our method introduces a principled distributional shift during decoding, enabling controlled exploration over an expanded latent manifold for diversification.

Related work explores guiding generation through the embedding space. For instance, SoftSRV trains full parametric embeddings to align outputs with a target distribution (DeSalvo et al., 2025). In contrast, our method adopts a hybrid design: combining continuous embeddings with token-level prompts, balancing the flexibility of embeddings with the interpretability of natural language.

We evaluate our approach on three distinct task domains that examine different aspects of diverse generation: (1) combinatorial task, based on a novel dataset we designed to evaluate the exploration effectiveness in large combinatorial solution spaces. Given a list of items, an LLM is prompted to generate multiple valid combinations whose values sum exactly to specified targets (Figure 1); (2) question generation (Gou et al., 2023; Rajpurkar et al., 2016), where LLMs are prompted to generate different questions that yield the same answer from a given context, testing diverse natural language generation; (3) molecule generation (Jang et al., 2025), which challenges domain-specific diversity. In particular, we work with a forward synthesis prediction task, where LLMs are required to generate multiple plausible products from fixed reagents and reactants, requiring chemical validity and structural diversity.

Our results demonstrate that soft prompt tuning can lead to more diverse responses while maintaining adherence to task constraints. The contributions of this work include:

- A lightweight, task-agnostic soft prompt diversification framework that enables controlled generation without fine-tuning or external training data. It can be directly applied to text diversification, showing consistent gains across various domains.
- A new target-sum combinatorial dataset featuring 50 real-life scenarios, each with an item list and a set of target values, evaluating LLMs' ability to generate diverse yet valid and optimal combinations.

## 2 BACKGROUND

**Diverse Text Generation** We define diverse text generation as the task of producing a diverse conditional distribution $p_\Theta(y|x)$, such that sampling from $p_\Theta$ yields semantically varied text outputs.

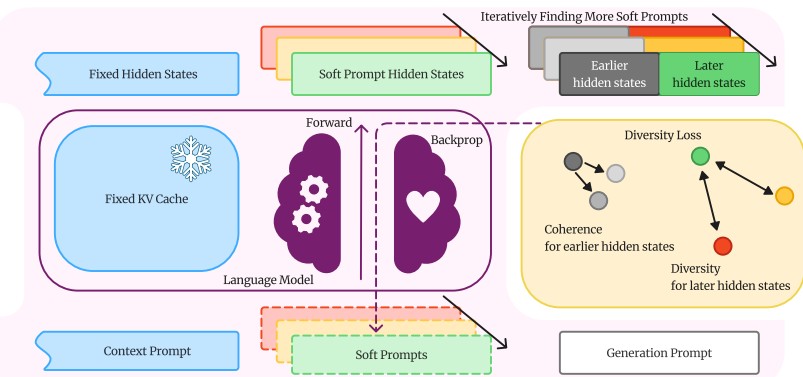

Figure 2: An illustration of our soft prompt tuning workflow.

Formally, given an input prompt $x_{1:L}$ and its generated continuation $y_{1:T}$, an autoregressive transformer model can be represented as:

$$p_\Theta(y|x) = \prod_{i=1}^{T} p_\Theta(y_i|x_{1:L}, y_{<i}).$$

At each token position $i$, the model computes hidden states as:

$$h_i = (h_i^{(1)}, \ldots, h_i^{(n)}) \in \mathbb{R}^{n \times d},$$

where $d$ is the embedding dimension and $h_i^{(j)}$ represents the output of transformer layer $j$. A causal attention mask ensures that $h_i^{(j)}$ attends only to previous hidden states $h_{<i}^{(\cdot)}$. Concatenating the input sequence and output generation to denote the full sequence as $z = (x, y)$, we have:

$$h_i = \text{LM}_\Theta(z_i, h_{<i}), \quad p_\Theta(y_{i+1}|x, y_{<i}) = \text{softmax}(W_\Theta h_i^{(n)}),$$

where $W_\Theta \in \mathbb{R}^{|\mathcal{V}| \times d}$ maps the final hidden state to vocabulary logits.

Since each next-token distribution is determined largely by the final-layer vector $h_i^{(n)}$, enhancing diversity in $p_\Theta$ can be reduced to the problem of increasing diversity in these last-layer hidden states.

**Soft Prompt** A soft prompt is a set of continuous learnable vectors $P \in \mathbb{R}^{n_p \times d}$ that is prepended to input embeddings to guide language model behavior, where $n_p$ denotes the prompt length and $d$ represents the embedding dimension. Unlike discrete text prompts, soft prompts operate directly in the embedding space, providing a parameter-efficient approach to model adaptation without modifying the model weights (Lester et al., 2021). These tunable embeddings participate in all attention computation and can effectively alter the model's hidden states and consequently the generated texts. Soft prompts have demonstrated significant performance benefits with minimal computational overhead across various tasks, making them particularly valuable for efficiently adapting large language models (Li & Liang, 2021).

## 3 SOFT PROMPT TUNING FOR DIVERSE TEXT GENERATION

The key intuition for our approach for promoting diversity is two-fold: first, we introduce a set of lightweight, continuous soft prompts that are sufficiently distinct from one another, which we prepend to a common input embedding. These prompts effectively induce shifts in the conditional token probability distribution, leading to diversified generations. Second, we optimize these soft prompts to steer the model's final hidden states into diverse regions of the representation space. Due to the autoregressive decoding nature of language models, these initial embedding differences propagate and amplify over subsequent decoding steps, yielding progressively more divergent textual continuations (Figure 2).

---

**Algorithm 1** Diverse Learning of Soft Prompts

---

**Input:** Context prompt embedding $E_c$, generation prompt embedding $E_g$, soft prompts $P_B \in \mathbb{R}^{n_p \times d}$, learning rate $\eta$, total epochs $T$, dynamic weight factor $\delta$, number of diverse tokens $m$
**Output:** Diversified soft prompts $P_B^*$

1:   $\mathcal{H} = \mathcal{M}(E_c, E_g)$
2:   $\ell = (h_{l-m}^g, .. h_l^g) \leftarrow$ last $m$ hidden states of $\mathcal{H}$
3:   $c = (h_{k+1}^g, ..., h_{l-m-1}^g) \leftarrow$ all but last $m$ hidden states of $\mathcal{H}$ in the generation prompt

4: **for** $t = 1$ **to** $T$ **do**
5:     $\widetilde{\mathcal{H}} = \mathcal{M}(E_c \oplus P_B \oplus E_g)$
6:     $\tilde{\ell} \leftarrow$ last $m$ hidden states of $\widetilde{\mathcal{H}}$
7:     $\tilde{c} \leftarrow$ all but last $m$ hidden states of $\widetilde{\mathcal{H}}$
8:     $d_{\text{last}} \leftarrow \| \tilde{\ell} - \ell \|_2$         ▷ 1. Last-$m$-token difference (to maximize)
9:     $d_{\text{ctrl}} \leftarrow \| \tilde{c} - c \|_2$         ▷ 2. Controlled difference (to minimize)
10:    $d_{\text{batch}} \leftarrow \left[ \frac{1}{B-1} \sum_{j=1, j \neq i}^{B} \| \tilde{\ell}_i - \tilde{\ell}_j \|_2 \right]_{i=1}^{B}$
11:                   ▷ 3. Average difference to other soft prompts in the batch (to maximize)
12:    **if** $t = 1$ **then**
13:       Store $d_0 \leftarrow d_{\text{ctrl}}$, set $w_c \leftarrow 0$
14:    **else**
15:       $\Delta \leftarrow \| d_{\text{ctrl}} - d_0 \|_2$
16:       $w_c \leftarrow \Delta / (\delta + \Delta)$
17:    **end if**

18:    $\mathcal{L} \leftarrow - (1 - w_c)(d_{\text{last}} + d_{\text{batch}}) + w_c \operatorname{mean}(d_{\text{ctrl}})$
19:    $P_B \leftarrow P_B - \eta \nabla_{P_B} \sum \mathcal{L}$         ▷ 4. Form loss and take gradient step
20: **end for**
   **return** $P_B$ as $P_B^*$

---

## 3.1 INITIALIZATION

To initialize diverse soft prompts, instead of direct sampling from the continuous embedding space $Z \in \mathbb{R}^d$, we construct a discrete space $\widetilde{Z} \in \mathbb{R}^d$ ($d$ denotes the embedding dimension) using scrambled Sobol sequences, which ensures a uniform coverage of the continuous space (Chi et al., 2005). We generate these Sobol sequences in a lower-dimensional space $d'$ where $d' \ll d$, then project them to the full embedding dimension using a matrix $A \in \mathbb{R}^{d \times d'}$ with values uniformly sampled from $(0, 1)$. This projection technique provides control over the magnitude of the resulting soft prompts, as larger values of $d'$ produce soft prompts with greater overall magnitude. This dimensional control introduces flexibility, as different tasks may benefit from soft prompts of varying magnitudes, allowing for efficient adaptation across diverse downstream applications (Lin et al., 2023).

## 3.2 SOFT PROMPT TUNING OBJECTIVE

Building on our initialization approach, we now turn to learning $r$ distinct soft prompts that could induce $r$ mutually diverse generations while maintaining task relevance.

We split the input into a context prompt and a generation prompt. The context prompt specifies the task requirements (e.g., the following is an example context prompt:"Context: The apple is red. Question: What is the color of the apple? Answer: Red. Your task is to generate a new question that can still be answered by the given answer based on the given context"), while the generation prompt elicits the response (e.g., "The new question is"). We denote their embeddings as $E_c$ and $E_g$.

Unlike standard approaches that prepend soft prompts at the beginning of the entire input (Li & Liang, 2021), we insert them between the context and generation prompts. This placement preserves the task instructions while allowing the soft prompts to influence generation trajectories.

The hidden states without and with soft prompts are denoted as ($\oplus$ denotes concatenation):

$$\mathcal{H} = \mathcal{M}(E_c \oplus E_g) = (h_1^c, h_2^c, \ldots, h_k^c, h_{k+1}^g, \ldots, h_l^g)$$

$$\widetilde{\mathcal{H}} = \mathcal{M}(E_c \oplus P \oplus E_g) = (\tilde{h}_1^c, \tilde{h}_2^c, \ldots, \tilde{h}_k^c, \tilde{h}_1^{sp}, \ldots, \tilde{h}_{n_p}^{sp}, \tilde{h}_{k+1}^g, \ldots, h_l^g)$$

Our loss function balances two objectives:

1. **Diversity:** Maximize the distance between the final $m$ hidden states produced with and without soft prompts, directly influencing the next token probability distribution. Additionally, we maximize pairwise distances between the final $m$ hidden states across different soft prompts to ensure mutually distinct generations.
2. **Consistency:** Minimize differences in the hidden state of earlier tokens in the generation prompt, preserving the semantic and task alignment with the original input context.

To ensure task alignment while promoting diversity, we employ a dynamic weighting mechanism to balance the two complementary objectives. This weight automatically adjusts based on how far the controlled token representations drift from their initial values. As training progresses, this mechanism prevents excessive deviation from task requirements while still encouraging diversity where intended. Finally, we use stochastic gradient descent (SGD) to update the soft prompts based on the computed loss. Given $r$ learned soft prompts, we choose a diversity-optimizing subset of size $q$ by minimizing the sum of pairwise cosine similarity, and prepend each selected prompt to the same generation prompt(s) to obtain $q$ distinct generations. Algorithm 1 provides the detailed implementations.

## 4 EXPERIMENTS

We evaluated our method across three distinct task domains. The combinatorial task challenges models to discover multiple distinct item subsets that sum precisely to target values. The question generation (QG) task (Rajpurkar et al., 2016) tests the ability to produce semantically varied questions yielding identical answers from given contexts. The forward synthesis molecule prediction tasks (FS-Mol) challenge models to predict plausible products from fixed reagents and reactants (Yu et al., 2024), using SMILES notation (Weininger, 1988) to represent molecular structures. In addition, we experiment with our methods on a different split of the question generation dataset and on a description-based molecule generation task. Details can be found in the Appendix H.

For each task, we decide two numbers for every input: $N_{\text{raw}}$, the number of candidate generations we produce, and $N_{\text{final}}$, the number we keep for evaluation. These values are chosen per task to reflect its difficulty and the size of its solution space, with details in Appendix B. Given an input, we first generate $N_{\text{raw}}$ candidates using the prompting templates in the Appendix C. For the soft prompt tuning framework specifically, we decide learning on $r = 5 * N_{\text{raw}}$ soft prompts and then select the most diverse $q = N_{\text{raw}}$ for generation. We then uniformly subsample $N_{\text{final}}$ candidates for evaluation. For the combinatorial task, we rank candidates by the absolute deviation between their sum and the target value and keep the top $N_{\text{final}}$. We apply the same selection procedures to all baselines and to our method to ensure a fair comparison.

### 4.1 EXPERIMENT SETUP

**Models and Hyperparameters** We implemented our approach using open-source language models tailored to each task domain. For combinatorial tasks and question generation, we utilized `Llama-3.1-8B-Instruct` (Dubey et al., 2024) due to its strong text understanding and generation capabilities. For molecule generation tasks, we employed a domain-specific model, `LlaSMol-Mistral-7B` (Yu et al., 2024) that is specifically fine-tuned on comprehensive chemical tasks to ensure accurate molecular representations. For optimal performance, we conducted a random search over several key hyperparameter configuration dimensions. This included soft prompt parameters (number of soft prompt tokens, intrinsic dimension, and number of tokens to diversify) and training parameters (learning rate, number of training epochs). The complete hyperparameter settings and optimal configurations for each task are detailed in the Appendix E.

**Baselines** We implemented three standard decoding strategies as our main baselines: Temperature Sampling, Nucleus Sampling, and Diverse Beam Search (Ackley et al., 1985; Holtzman et al.,

2019; Vijayakumar et al., 2018). We systematically evaluated each strategy across multiple hyperparameter configurations to ensure a comprehensive comparison. For Temperature Sampling, we tested temperatures $\in \{0.6, 0.8, 1.0, 1.2, 1.4\}$ while fixing top-$p = 0.95$. For Nucleus Sampling, we tested top-$p \in \{0.8, 0.85, 0.9, 0.95, 1.0\}$ while maintaining temperature $= 1.0$. For Diverse Beam Search, we explored beam group sizes $\in \{1, 2\}$ and diversity penalties $\in \{0.6, 0.8, 1.0\}$, totaling 6 configurations. For each baseline method with each specific hyperparameter setting, we ran our proposed method using the identical decoding strategy and hyperparameters. This ensures that any performance differences can be attributed to our method rather than variations in the underlying decoding configuration.

Due to the extensive hyperparameter experiments, we initially evaluated all configurations on approximately 10% of the dataset with 3 independent runs per configuration, except for the Diverse Beam Search due to its deterministic nature. We then selected the best-performing hyperparameter setting from each baseline method and conducted full dataset evaluations for both the baselines and our proposed method under matching configurations. The findings on the full dataset closely matched those on the subset (see Appendix H for complete results). In the following sections, we present averaged performance across the 3 independent runs for clarity. Individual run results and standard deviations are also provided in Appendix H. Additionally, we benchmarked our method against GPT-4o-mini using optimized diversity parameters (temperature = 1.2, presence penalty = 0.8) (Achiam et al., 2023), with full results available in the Appendix as well.

## 4.2 COMBINATORIAL GENERATION TASK

**Dataset and Metrics** We created a novel combinatorial dataset by prompting GPT-4o to generate 50 realistic scenarios spanning diverse domains. For each scenario, we defined 20 distinct target values and let GPT-4o generate a contextually appropriate list of 30 items with associated values. We implemented verification protocols to ensure that each target value could be achieved through at least 20 different combinations, yielding 1,000 combinatorial problems for evaluation. Each target value has an average of 1,624,205 possible valid solutions. More details about the dataset construction can be found in the Appendix G.

We evaluate solution quality using Mean Relative Sum Error (MRSE), which quantifies the average absolute deviation of each combination's sum from the target, normalized by the target itself, offering an overall sense of how closely the model's outputs approximate the desired total. For diversity evaluation, we employ Uniqueness, which calculates the proportion of distinct combinations.

**Results** Compared with temperature and nucleus sampling, our method achieves more diverse responses while improving quality, representing a clear Pareto improvement as shown in Figure 3(a). Averaged over temperature sampling baselines, our method decreases MRSE from 0.636 to 0.536 ($-0.1$; $-15.7\%$) while increasing Uniqueness from 95.39% to 97.46% ($+2.07$; $+2.2\%$). For top-$p$ sampling, MRSE drops from 0.616 to 0.506 ($-0.110$; $-17.9\%$) with Uniqueness rising from 95.66% to 97.47% ($+1.81$; $+1.9\%$) when shifting to our method. Meanwhile, diverse beam search proves uncompetitive on this combinatorial task, achieving only 50–78% Uniqueness and higher MRSE, while sampling methods already attain 95–98% Uniqueness baselines that our approach further enhances. The quality–diversity balance achieved on these complex problems demonstrates our framework's effectiveness at navigating diverse generations in high-dimensional solution spaces.

This task further emphasizes the importance of diverse generations: While enhanced diversity has the possibility of coming at the cost of decreased quality, it can actually lead to better solutions by more thoroughly exploring the solution space. Furthermore, standard language models like GPT-4o-mini exhibit limited diversity (93.47%) (Table 12), failing to explore the rich solution spaces characteristic of real-world problems. This underscores both the challenge posed by our synthetic combinatorial dataset and the practical value of our approach for complex generation tasks.

## 4.3 QUESTION GENERATION

**Dataset and Metrics** We evaluated our approach using the SQuAD test datasets (Rajpurkar et al., 2016), employing SQuAD 1 data split established in prior work by Zhou et al. (2017). For a comprehensive quality assessment, we employ the QGEval framework (Fu et al., 2024) with seven interpretable criteria: fluency, clarity, conciseness, relevance, consistency, answerability, and answer consistency. We implement this evaluation using UniEval (Zhong et al., 2022), a T5-based system

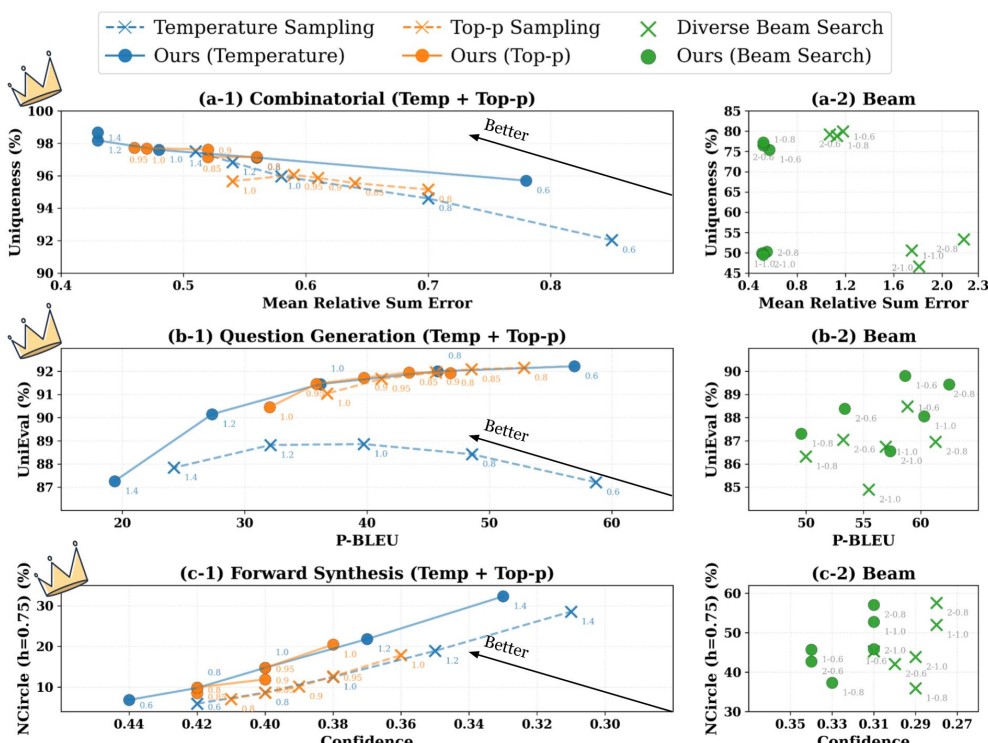

Figure 3: Quality-Diversity Trade-off across tasks. Each marker represents an independent experiment with a distinct hyperparameter configuration (temperature, top-p, beam search) for baselines and our method. The annotation near each marker indicates the specific hyperparameter setting: temperature for Temperature Sampling, top-$p$ for Nucleus Sampling, beam group size-diversity penalty for Diverse Beam Search. Markers in the top-left corner indicate better performance.

that frames assessment as a binary question-answering task (e.g.,"Is this question fluent") and derives normalized scores from yes/no probabilities, which have a strong correlation with human judgments. The UniEval score presented is the score averaged over the the seven dimensions. For diversity assessment, we first calculate the next-token prediction loss when generating the target answer from the generated question using `Llama-3.1-8B-Instruct` as our QA reward model. Then we filter out low-quality generations by removing questions whose answer-prediction loss exceeds 1.0. We then measure diversity among the remaining questions using Pairwise-BLEU (Self-BLEU), calculating the sentence-level metrics of each question against all others as references.

**Results** In the Question Generation task, we observe the expected trade-off between diversity and quality as lower P-BLEU (higher diversity) correlates with lower UniEval score (lower quality) in Figure 3(b). Nonetheless, our method consistently outperforms all baselines by shifting the diversity-quality curve upward and leftward, enabling simultaneous improvements in both dimensions. Compared to temperature sampling baselines (blue "×" markers), our approach achieves substantial gains in quality: 3-5% higher UniEval scores while improving diversity by 2 to 4 points on the Pairwise-BLEU metric. With respect to nucleus sampling, both methods maintain high UniEval scores as the top-$p$ value increases, but our method achieves meaningfully superior diversity at matching top-$p$ values. Specifically, our method reduces Pairwise-BLEU to 46.8 versus 52.8 for top-$p$=0.8, and to 32.0 versus 36.7 for top-$p$=1.0, indicating substantially more diverse outputs without sacrificing quality. In contrast, diverse beam search remains constrained to low-diversity regions and provides only marginal quality improvements.

### 4.4 MOLECULE GENERATION

**Dataset and Metrics** For the forward synthesis prediction task (FS-Mol), we utilize the test set from Yu et al. (2024), which consists of cleaned and curated reactions from the USPTO-full chemi-

cal reaction dataset (Lowe, 2017). To evaluate the quality of the generated molecules, we use RXN-Mapper confidence (Schwaller et al., 2021), which measures the model's certainty in atom-mapping predictions. In practice, a higher confidence value indicates that atom-mapping was driven by clear attention signals, reflecting the model's greater certainty in the forward synthesis outcome. For the diversity metric, we use NCircles$_h$ (Xie et al., 2023), which measures the largest subset where no two molecules have Tanimoto similarity (Bajusz et al., 2015) above threshold $h$. We then normalize NCircles$_h$ as a percentage of total generations to enable consistent cross-task comparisons.

**Results**  As shown in Figure 3(c), diverse beam search achieves extremely high diversity in the molecule generation task, producing roughly twice the NCircle values of baseline methods. However, this comes at the cost of substantial quality degradation. In contrast, our method enhances diversity while maintaining or even improving quality. Averaged across temperature settings, our approach increases confidence from 0.372 to 0.392 (+0.020; +5.4%) while NCircle rises from 14.92 to 17.11 (+2.19; +14.7%). Under nucleus sampling, we observe similar patterns: confidence improves from 0.388 to 0.404 (+0.016; +4.1%) with NCircle increasing from 11.26 to 13.11 (+1.85; +16.4%). These results demonstrate that while diverse beam search maximizes diversity at the expense of severe quality loss, our method provides a more balanced approach, achieving effective diversification with a superior quality–diversity trade-off. Furthermore, our method exhibits greater robustness to increased randomness in sampling strategies. As temperature and top-p values increase, the performance gap between our method ("○" markers) and baselines ("×" markers) widens, indicating better preservation of quality metrics under higher stochasticity.

## 5 DISCUSSION

**Diversity requires guided exploration, not random perturbation.**  We compared our method against Hidden State Noise Injection (HSNI) experiment, where we add Gaussian noise to the final $m$ token hidden states with magnitude normalized by state norms. As shown in Table 1, HSNI fails to produce diverse outputs in Combinatorial and Question Generation tasks, with very similar diversity metrics to temperature sampling baseline. This demonstrates that we need meaningful perturbations in the large latent space for diversification. Although HSNI can produce diverse responses in FS-Mol with a significant increase in NCircle value, quality deteriorates a lot, further demonstrating that different subspaces of hidden states may serve distinct functions such as task adherence and diversification. Random perturbation affects both indiscriminately, whereas our soft prompt approach provides targeted influence in directions that preserve task constraints while encouraging meaningful diversification. The failure of random noise highlights the need for enhanced understanding of the structure of the hidden state space rather than relying on undirected exploration. Full results with complete metrics across all tasks can be found in the Appendix H.

| Method | Combinatorial | | QG (SQuAD 1) | | FS-Mol | |
|---|---|---|---|---|---|---|
| | MRSE ↓ | Unique (%) ↑ | UniEval (%) ↑ | P. BLEU ↓ | Confidence ↑ | NCircle(%) ↑ |
| Temperature | 0.53 | 97.76 | **91.09** | 33.16 | **0.36** | 19 |
| Ours | **0.44** | **98.53** | 90.12 | **27.4** | **0.36** | 21.42 |
| HSNI | 0.52 | 97.99 | 91.06 | 33.19 | 0.33 | **26.01** |

Table 1: Comparison of our method with Hidden-State Noise Injection (HSNI) and temperature sampling across three tasks. Temperature remains the same as 1.2 across three methods.

**One set of soft prompts might not fit all.**  The current training framework requires task-specific soft prompts for effective diversification, which limits its generalizability. Nonetheless, instead of generic approaches, each task might benefit from tailored strategies. The intuition lies in diversification isn't just about one unified strategy: Variety, it's about using strategies tailored to each task. In combinatorial settings, models should generate distinct, valid combinations adhering to numerical constraints, not random permutations. In question generation, the focus should be context-aware, answerable questions rather than simple paraphrases. True diversification must align with the structure and goals of each specific task. To validate this hypothesis, we analyzed trained soft prompts across tasks by computing average L-2 distances between prompts trained for different tasks from identical initializations. The results in Table 9 show that diagonal values are near-zero, confirming stable training directions within tasks, while similar tasks (e.g., SQuAD 1 and SQuAD 2) maintain

close soft prompt representations. Moreover, dissimilar tasks exhibit significant prompt divergence, reinforcing that different tasks might require distinct directions in the embedding space for optimal diversification.

**Limitations**    While our soft prompt tuning approach offers significant advantages for diverse text generation, it has several limitations. First, our method's strategic placement of soft prompts in the input embeddings relies on continuous hidden state propagation through an autoregressive sequence, limiting its direct applicability to other model architectures such as encoder-decoder models. Second, the continuous nature of soft prompts inherently limits interpretability, making it challenging to precisely understand how specific prompt vectors influence generation trajectories. Moreover, our current implementation does not explicitly incorporate reasoning paths in language models, while the soft prompts may alter these reasoning processes in ways that contribute to output diversity, but this interaction remains unexplored. Future work could extend this approach to non-autoregressive architectures, developing techniques to improve soft prompt interpretability, and understanding how soft prompts could have an impact on the reasoning pathways.

Finally, the synthetic combinatorial dataset may not fully capture real-world complexity. Initial attempts to introduce additional constraints such as limiting selected items (e.g., requiring item value sums to equal $N$ while total items $\leq M$ proved challenging, as LLM-generated item lists under stricter constraints often lacked sufficient valid solutions, which is particularly problematic for diversity-focused tasks requiring rich solution spaces. Since combinatorial problems remain both common and challenging for LLMs, constructing more rigorous datasets through combined LLM generation and algorithmic validation represents an important direction for future work.

## 6 RELATED WORKS

**Training-Free Diverse Text Generation**    Most training-free diverse text generation approaches focus on manipulating probability distributions during decoding. Methods such as diverse beam search (Vijayakumar et al., 2018), nucleus sampling (Holtzman et al., 2019), top-k sampling (Fan et al., 2018) and minimum-probability sampling (Minh et al., 2024) have been widely adopted to enhance output diversity without requiring model fine-tuning. Entropy-guided approaches like $\eta$-sampling and microstat sampling offer another direction by dynamically modifying the candidate token pool based on the entropy of the token distribution (Hewitt et al., 2022; Basu et al., 2020). For combinatorial optimization specifically, LMEA (Liu et al., 2024) leverages large language models as evolutionary search operators over discrete solution spaces without gradient-based training. Our method extends beyond these approaches by operating directly in the embedding space, enabling a more precise control over the diversity-quality balance.

**Fine-tune based Diverse Text Generation**    To encourage diverse text outputs, a range of fine-tuning strategies has been explored. In encoder–decoder models (Cho et al., 2014), mixture-of-decoders frameworks have been devised to produce multiple hypotheses per input (He et al., 2018; Shen et al., 2019). Inverse reinforcement learning has also been employed to learn reward functions that explicitly promote diversity (Shi et al., 2018). More recently, generative flow networks have recast autoregressive generation as flows over a DAG of partial states, sampling complete outputs in proportion to a user-defined reward (Bengio et al., 2021). While these techniques achieve strong diversity, they typically depend on carefully crafted rewards and extensive task-specific fine-tuning. In contrast, our soft-prompt tuning framework delivers comparable diversity through lightweight, parameter-efficient optimization that can be applied broadly across domains.

## 7 CONCLUSIONS

In this work, we introduced a lightweight diverse text generation framework using soft prompt tuning that achieves high output diversity while maintaining task constraint adherence. By optimizing these continuous prompts, our approach induces targeted distributional shifts guiding language models toward diverse outputs. Experiments across combinatorial tasks, question generation, and molecular generation have demonstrated superior diversity while maintaining or even increasing generation quality. In particular, the framework shows exceptional performance in domains with expansive solution spaces, particularly in our proposed synthetic combinatorial dataset. Future work could explore adapting this approach to additional model architectures, improving soft prompt interpretability, and investigating soft prompt effects on reasoning pathways.

**Ethics Statement** This approach potentially broadens the application of language models by enabling diverse generation. It may exaggerate the hallucination of language models, so the generated information should be more carefully examined.

**Reproducibility statement** Models used in this research such as Llama-3.1-8B-Instruct, are open-sourced with citation provided in the main text when mentioned. Code and the dataset will be published as an open-source repository on GitHub after the anonymous review period.

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

## A USE OF LLM STATEMENT

Large Language Models were used only as writing and visualization aids, such as improving the clarity of text and helping format tables/figures in LaTeX. All research ideas, implementations, analyses, and conclusions are entirely the authors' own.

## B DATASET STATISTICS

| Task | Size | Subset Size | $N_{raw}$ | $N_{final}$ |
|---|---|---|---|---|
| QG (SQuAD1) | 8,964 | 1000 | 20 | 5 |
| QG (SQuAD2) | 11,877 | 1000 | 20 | 5 |
| Desc-Mol | 1,060 | 100 | 100 | 50 |
| FS-Mol | 4,062 | 500 | 100 | 50 |
| Combinatorial | 1,000 | 100 | 50 | 20 |

Table 2: Datasets and per-input generation settings. $N_{raw}$: candidates produced per input; $N_{final}$: candidates retained for evaluation.

## C CONTEXT AND GENERATION PROMPT BASED ON TASKS

| Task | Generation Prompt |
|---|---|
| QG | `New Question:` |
| Desc-Mol | `New SMILES: <SMILES>` |
| FS-Mol | `Product SMILES: <SMILES>` |
| Combinatorial | `Selection: <SELECT>` |

Table 3: Generation prompts that are prepended by soft prompts

| Task | Prompt Template |
|------|-----------------|
| QG | Your task is to generate a new question.  The new question should still be answered correctly using the same answer.  The new question should be relevant to the context.

***Start of an example***
Context:  Antarctica is the driest continent on Earth, receiving less precipitation than the Sahara Desert.
Example Question:  Which continent receives less precipitation than the Sahara?
Answer:  Antarctica
New Question:  Which continent is the driest on Earth?
Answer:  Antarctica
***End of an example***

***Your turn***
Context:  {context}
Example Question:  {question}
Answer:  {answer} |
| Desc-Mol | Your task is to generate a molecule based on the description.  Your output should be a SMILES string.

***Start of an example***
Description:  The molecule is a member of the class of tripyrroles ...  ring assembly.
New SMILES: <SMILES> CCCCC1=C(C)NC(/C=C2\N=C(C3=CC=CN3)C=C2OC)=C1 </SMILES>
***End of an example***

***Your turn***
Description:  {description} |

Table 4: Context prompt templates for question generation task (QG) and description guided molecule generation task (Desc-Mol)

| Task | Prompt Template |
|---|---|
| FS-Mol | Your task is to predict the product of a chemical reaction. Given the SMILES of reactants and reagents, generate the SMILES of the most likely product. Your output should be a SMILES string.

\*\*\*Start of an example\*\*\*
Reactants and Reagents: \<SMILES\> NC1=CC=C2OCOC2=C1.O=CO \</SMILES\>
Product SMILES: \<SMILES\> O=CNC1=CC=C2OCOC2=C1 \</SMILES\>
\*\*\*End of an example\*\*\*

\*\*\*Your turn\*\*\*
Reactants and Reagents: \<SMILES\> {input} \</SMILES\> |
| Combinatorial | Given a target value and a list of items with their values, your task is to select a subset of items such that the sum of their values is as close as possible to the target value. The specifc meaning of "item value" and "target value" may vary from task to task, but the general goal is the same: select a list of items whose value sum up to the target.
Make sure to first output your answer in \<SELECT\> \</SELECT\> tags. The answer format should be a list of item IDs, separated by commas.

\*\*\*Start of an example\*\*\*
Task: Create a box of chocolates with a total weight of 200g.
Item list: [{"id": 1, "name": "Chocolate A", "value": 100 }, ...]
Selection: \<SELECT\> [1,3,4] \</SELECT\>.
\*\*\*End of an example\*\*\*

\*\*\*Your turn\*\*\*
Task: {task}
Item list: {item list} |

Table 5: Context prompt templates for forward synthesis molecule generation (FS-Mol) and Combinatorial tasks.

# D    Sample Generation Results

## D.1    Question Generation

| SQuAD 1 |
|---|
| **Input:** In much of the U.S. South, many white students migrated to the academies, while public schools became in turn more heavily concentrated with African-American students (see List of private schools in Mississippi). |
| **Gold Question:** School desegregation in the United States led to an increased number of students of what ethnicity in public schools? |
| **Answer:** African-American |
| **GPT-4o-mini:**
  1. School desegregation in the United States led to a decrease in the number of students from which racial group in traditional private schools?
  2. Following school desegregation in the United States, what ethnic group became predominant in schools previously known as public schools in the U.S. South?
  3. School desegregation in the United States led to schools becoming predominantly attended by what group?
  4. What demographic shift did school desegregation in the United States lead to in public schools in the U.S. South?
  5. At public schools in the US South after desegregation, students of what ethnicity became concentrated in greater numbers?
**Ours:**
  1. Which group of students became more prevalent in public schools as a result of school desegregation in the United States?
  2. Which demographic saw a rise in enrollment in public schools as white students moved to private academies in the U.S. South?
  3. What demographic shift occurred in public schools as a result of school desegregation in the United States?
  4. What ethnicity notably became more prevalent in public schools due to school desegregation in the U.S. South?
  5. What ethnicity saw a significant rise in enrollment in public schools due to school desegregation in the United States? |
| **Input:** Deke Slayton, the grounded Mercury astronaut who became Director of Flight Crew Operations for the Gemini and Apollo programs, selected the first Apollo crew in January 1966, with Grissom as Command Pilot, White as Senior Pilot, and rookie Donn F. Eisele as Pilot. |
| **Gold Question:** Who selected the first Apollo crew? |
| **GPT-4o-mini:**
  1. Who appointed the crew for the first Apollo mission?
  2. Who was responsible for choosing the initial crew of the Apollo program?
  3. Who was responsible for choosing the crew for the first Apollo mission?
  4. Who was responsible for choosing the initial crew of the Apollo mission?
  5. Who was responsible for selecting the first crew of the Apollo program?
**Ours:**
  1. Whose team chose the first Apollo crew?
  2. Who selected the original Apollo crew?
  3. Who became Director of Flight Crew Operations for the Gemini and Apollo programs and later selected the first Apollo crew?
  4. What astronaut selected the first crew for the Apollo mission?
  5. Who chose the initial crew for the Apollo space missions? |
| **Input:** This mechanism is still the leading theory today; however, a second theory suggests that most cpDNA is actually linear and replicates through homologous recombination. |
| **Gold Question:** How does the secondary theory say most cpDNA replicates? |
| **GPT-4o-mini:**
  1. What process does the secondary theory propose for the replication of most cpDNA?
  2. What process is proposed by the secondary theory as a method for cpDNA replication?
  3. What replication process does the secondary theory propose for the majority of cpDNA?
  4. What method does the secondary theory propose for the replication of most cpDNA?
  5. What method does the second theory propose for the replication of most cpDNA?
**Ours:**
  1. What replication process does the secondary theory propose for most cpDNA?
  2. In what way does the secondary theory propose cpDNA replicates?
  3. According to the secondary theory, what does cpDNA replicate through?
  4. According to the secondary theory, in what mechanism do cpDNA usually replicate?
  5. According to the secondary theory, what is the mode of replication of most cpDNA? |

Table 6: Examples of question generation results from SQuAD 1 split. For each input context, we show the original gold question and the top 5 generated questions from GPT-4o-mini and our method. Our questions demonstrate greater lexical and structural diversity while remaining faithful to the context.

# E    Soft Prompt and Training Hyperparameters

We perform a 15-trial random search for each task over soft prompt hyperparameters and training hyperparameters during diverse learning. The specific search range for each hyperparameter can be found in Table 7, and the resulting optimal hyperparameter settings for each task are detailed in Table 8

| Hyperparameter | Search Space |
|---|---|
| # soft prompt tokens ($n_p$) | {1, 3, 5, 10} |
| Last $m$ tokens to diversify | {1, 3} |
| Intrinsic dimension ($d'$) | {10, 50, 100, 500, 1000} |
| Learning rate ($\eta$) | {1e-2, 1e-3, 1e-4} |
| Training Epoch ($T$) | {20, 30, 50} |
| Dynamic weight ($\delta$) | {0, 30, 100} |

Table 7: Hyperparameter search space.

| | QG | Desc-Mol | FS-Mol | Combinatorial |
|---|---|---|---|---|
| # soft prompt tokens ($n_p$) | 5 | 1 | 5 | 5 |
| Last $m$ tokens to diversify | 1 | 1 | 1 | 1 |
| Intrinsic dimension ($d'$) | 50 | 1000 | 50 | 50 |
| Learning rate ($\eta$) | 1e-4 | 1e-4 | 1e-4 | 1e-4 |
| Training Epoch ($T$) | 20 | 20 | 20 | 20 |
| Dynamic weight ($\delta$) | 100 | 100 | 100 | 100 |

Table 8: Best hyperparameter settings for diverse generation tasks through random search.

## F  SOFT PROMPT GENERALIZABILITY

| Task | Combinatorial | SQuAD 1 | SQuAD 2 | Desc-Mol | FS-Mol |
|---|---|---|---|---|---|
| Combinatorial | 0.001 | 1217.23 | 1217.23 | 1224.91 | 450.40 |
| SQuAD 1 | - | 1.70e-5 | 0.003 | 744.61 | 1186.44 |
| SQuAD 2 | - | - | 9.08e-5 | 744.61 | 1186.45 |
| Desc-Mol | - | - | - | 1.08e-5 | 1194.86 |
| FS-Mol | - | - | - | - | 0.002 |

Table 9: Inter-Task Soft Prompt L-2 Distance. Values indicate the L-2 Distance between trained soft prompt learning on different task domains.

## G  DATASET CONSTRUCTION DETAILS

The section introduced more details about dataset construction. The dataset generation involved the following three steps:

1. Combinatorial task scenario generation and target value specification
2. Item list generation
3. Solution verification

First, we prompted GPT-4o to generate a list of realistic combinatorial task scenarios. Each scenario includes three components: a scenario name, a description, and a set of 20 target values. To ensure consistency and relevance, we provided an example in the prompt that reflects our desired "realistic knapsack" style tasks:

*"scenario name":"Food & Nutrition", "description": "Create a meal exactly totaling {} calories", "target values": [300, ..., 2000]*

We further manually reviewed and filtered the generated scenarios to ensure quality and diversity, ultimately selecting around 80 candidate scenarios. Each entry has a format similar to the one above. Next, for each accepted scenario, we crafted a system prompt to generate the corresponding item list. This second step again used GPT-4o, with the following prompt format:

You are a data generator for combinatorial optimization tasks.

Given the following task description: `"Create a meal exactly totaling N calories"`, where $N \in [300, \ldots, 2000]$ is the target value.

Generate a list of 30 unique items appropriate to the `"Food & Nutrition"` scenario. Each item should include:

- `id` (starting from 1),
- `name` (realistic item names),
- `value` (a number appropriate to the task's unit, within a sensible range).

Your output should be in JSON Lines (`.jsonl`) format, with one JSON object per line.

Finally, we implemented a verification step using a dynamic programming algorithm to exhaustively enumerate valid combinations (without replacement) of items that sum to each target value. Any scenario in which fewer than 20 valid combinations existed for any target value was discarded. After this filtering step, we kept 50 high-quality scenarios for the final dataset.

## H  COMPREHENSIVE RESULTS

In this section, we present detailed results for the three tasks discussed in the main text, along with two additional tasks. Each subsection contains three primary results tables. The first two tables compare our method against standard decoding baselines, with results averaged over three independent runs on a smaller dataset (diverse beam search was run once due to its deterministic nature). The second table presents results on a larger dataset, comparing our method against standard decoding baselines with optimized hyperparameters, a GPT-4o-mini baseline, a hidden states Gaussian noise injection baseline, and an ablation of our method using random initialization instead of Sobol sequence-based initialization for soft prompts.

For simplicity, in the results table, we use "tem 0.6" stands for temperature=0.6 when running temperature sampling baseline (top-$p$=0.95); "topp 0.8" stands for top-$p$=0.8 when running top-$p$ sampling baseline (temperature=1.0); "beam 1 penalty 0.6" stands for beam group size is 1 and diversity penalty is 0.6.

Aside from the task-specific diversity metric we presented in the main text, we employ a unified approach to measuring diversity across all tasks through the Vendi Score (Friedman & Dieng, 2022). This metric is calculated as the exponential of the Shannon entropy of the eigenvalues derived from the samples' normalized similarity matrix. The Vendi Score offers high interpretability, where a score of $m$ indicates that the evaluated set exhibits $m$ unique items. For consistent cross-task interpretation, we express the Vendi Score as a percentage of total generations, creating a normalized diversity metric independent of generation count. For our text-based evaluations, we construct an $N \times N$ similarity matrix, with each cell representing the dot-product similarity between the embeddings of two generated outputs. To ensure high-quality semantic representations, we utilize the `stella_en_1.5B_v5` embedding model (Zhang et al., 2024) for all Vendi Score calculations.

### H.1  COMBINATORIAL TASK

In addition to MRSE, we evaluate solution quality using another two metrics: Within-20% Acceptance Rate, Within-50% Acceptance Rate. The acceptance rates measure the proportion of unique combinations whose summed values fall within 20% and 50% of the target value, respectively, providing a metric of near-miss accuracy. Comprehensive results can be found in Table 10-12.

### H.2  QUESTION GENERATION TASK (SQUAD1)

In addition to the metrics introduced in the main text, we evaluate the generation quality using Oracle-BLEU, which measures the highest BLEU-4 between the target question and any generated question. Comprehensive results can be found in Table 13-15.

| Hyperparameters | Quality Metrics | | | Diversity Metrics | |
|---|---|---|---|---|---|
| | 20% Acpt ↑ | 50% Acpt ↑ | MRSE ↓ | Unique (%) ↑ | Vendi (%) ↑ |
| tem 0.6 | 25.33 (1.07) | 49.72 (0.93) | 0.85 (0.02) | 92.03 (0.37) | 10.13 (0.18) |
| tem 0.8 | 26.98 (0.48) | 57.38 (0.41) | 0.70 (0.01) | 94.60 (0.51) | 12.54 (0.30) |
| tem 1.0 | 26.38 (0.45) | 61.62 (0.77) | 0.58 (0.02) | 95.97 (0.23) | 14.57 (0.33) |
| tem 1.2 | 25.97 (0.28) | 62.02 (0.98) | 0.54 (0.02) | 96.83 (0.31) | 16.05 (0.27) |
| tem 1.4 | 25.50 (0.83) | 62.18 (1.49) | 0.51 (0.01) | 97.50 (0.14) | 16.52 (0.12) |
| topp 0.8 | 26.05 (0.40) | 55.08 (0.90) | 0.70 (0.01) | 95.15 (0.46) | 13.39 (0.17) |
| topp 0.85 | 26.43 (0.84) | 58.17 (1.04) | 0.64 (0.01) | 95.55 (0.20) | 13.76 (0.22) |
| topp 0.9 | 25.57 (1.01) | 59.58 (1.04) | 0.61 (0.01) | 95.87 (0.34) | 14.28 (0.14) |
| topp 0.95 | 25.32 (0.96) | 60.90 (0.51) | 0.59 (0.00) | 96.05 (0.07) | 14.46 (0.12) |
| topp 1.0 | 26.12 (0.91) | 61.23 (0.91) | 0.54 (0.00) | 95.67 (0.16) | 14.67 (0.16) |
| beam 1 penalty 0.6 | 11.90 | 30.00 | 1.18 | 79.95 | 11.74 |
| beam 1 penalty 0.8 | 12.30 | 30.30 | 1.07 | 79.20 | 11.84 |
| beam 1 penalty 1.0 | 12.45 | 28.75 | 1.13 | 78.80 | 12.10 |
| beam 2 penalty 0.6 | 6.65 | 15.00 | 2.18 | 53.30 | 9.60 |
| beam 2 penalty 0.8 | 7.25 | 16.00 | 1.75 | 50.60 | 9.45 |
| beam 2 penalty 1.0 | 5.90 | 13.80 | 1.81 | 46.65 | 9.48 |

Table 10: Combinatorial: Standard decoding baseline results

| Hyperparameters | Quality Metrics | | | Diversity Metrics | |
|---|---|---|---|---|---|
| | 20% Acpt ↑ | 50% Acpt ↑ | MRSE ↓ | Unique (%) ↑ | Vendi (%) ↑ |
| tem 0.6 | 22.27 (0.39) | 49.63 (0.90) | 0.78 (0.02) | 95.70 (0.22) | 12.34 (0.10) |
| tem 0.8 | 27.48 (0.66) | 60.97 (0.64) | 0.56 (0.01) | 97.12 (0.23) | 14.18 (0.03) |
| tem 1.0 | 28.47 (1.01) | 65.80 (1.29) | 0.48 (0.01) | 97.60 (0.39) | 16.05 (0.13) |
| tem 1.2 | 28.20 (0.44) | 66.57 (0.40) | 0.43 (0.01) | 98.18 (0.06) | 17.51 (0.22) |
| tem 1.4 | 26.92 (0.46) | 64.72 (0.39) | 0.43 (0.01) | 98.68 (0.24) | 17.77 (0.18) |
| topp 0.8 | 29.32 (0.76) | 61.30 (0.76) | 0.56 (0.01) | 97.17 (0.09) | 14.69 (0.08) |
| topp 0.85 | 28.50 (0.52) | 62.45 (1.26) | 0.52 (0.01) | 97.15 (0.23) | 15.21 (0.12) |
| topp 0.9 | 28.12 (0.47) | 62.92 (0.63) | 0.52 (0.00) | 97.62 (0.22) | 15.62 (0.24) |
| topp 0.95 | 29.98 (0.78) | 67.03 (1.02) | 0.46 (0.01) | 97.72 (0.02) | 16.02 (0.13) |
| topp 1.0 | 26.67 (1.20) | 64.30 (0.74) | 0.47 (0.01) | 97.68 (0.19) | 16.18 (0.17) |
| beam 1 penalty 0.6 | 18.50 | 42.35 | 0.57 | 75.40 | 11.61 |
| beam 1 penalty 0.8 | 19.75 | 44.65 | 0.52 | 76.50 | 11.76 |
| beam 1 penalty 1.0 | 17.95 | 42.90 | 0.52 | 77.20 | 12.09 |
| beam 2 penalty 0.6 | 12.10 | 28.90 | 0.55 | 50.30 | 10.73 |
| beam 2 penalty 0.8 | 12.85 | 29.70 | 0.51 | 49.90 | 11.05 |
| beam 2 penalty 1.0 | 11.75 | 29.35 | 0.52 | 49.55 | 11.16 |

Table 11: Combinatorial: Soft prompt tuning results, to be compared with the standard decoding baseline

### H.3 QUESTION GENERATION TASK (SQUAD2)

SQuAD2 is another data split based on the SQuAD test datasets (Rajpurkar et al., 2016)Du et al. (2017). Comprehensive results can be found in Table 16-18

### H.4 DESCRIPTION-GUIDED MOLECULE GENERATION (DESC-MOL)

For description-guided molecule generation, we adapted the dataset from Fang et al. (2024), originally comprising 331,261 SELFIES-description molecule pairs extracted from PubChem (Kim et al., 2020). We first converted all SELFIES to SMILES notation and implemented a two-stage filtering process: (1) removing descriptions paired with only a single molecule to ensure the possibility for diversity. (2) eliminating overly specific descriptions or those referencing existing compounds. This curation process yielded 1,060 high-quality descriptions primarily characterizing broader chemical families with sufficient structural flexibility for novel generation.

| Method | Quality Metrics | | | Diversity Metrics | |
|---|---|---|---|---|---|
| | 20% Acpt ↑ | 50% Acpt ↑ | MRSE ↓ | Unique (%) ↑ | Vendi (%) ↑ |
| GPT-4o-mini | 39.37 | 74.91 | 0.34 | 93.47 | 4.50 |
| Temperature Sampling | 27.40 | 64.31 | 0.53 | 97.67 | 16.21 |
| Ours (temp) | 28.06 | 67.42 | 0.44 | 98.53 | 17.13 |
| Top-$p$ Sampling | 27.90 | 63.60 | 0.54 | 97.13 | 15.01 |
| Ours (top-$p$) | 27.46 | 65.29 | 0.48 | 98.18 | 16.20 |
| Diverse Beam Search | 12.16 | 30.04 | 1.12 | 79.17 | 11.78 |
| Ours (beam) | 19.50 | 45.00 | 0.50 | 74.64 | 11.72 |
| Ours | 28.06 | 67.42 | 0.44 | 98.53 | 17.13 |
| Randomly initialized soft prompt | 32.96 | 71.69 | 0.42 | 98.63 | 16.29 |
| Hidden state noise injection | 28.37 | 65.43 | 0.52 | 97.99 | 16.00 |

Table 12: Combinatorial: Comparison between soft prompt tuning method with other baselines

| Hyperparameters | Quality Metrics | | Diversity Metrics | |
|---|---|---|---|---|
| | O. BLEU ↑ | UniEval (%) ↑ | P. BLEU ↓ | Vendi (%) ↑ |
| tem 0.6 | 31.76 (0.01) | 87.21 (0.11) | 58.70 (0.16) | 37.63 (0.16) |
| tem 0.8 | 31.07 (0.11) | 88.42 (0.14) | 48.58 (0.85) | 39.99 (0.24) |
| tem 1.0 | 29.51 (0.48) | 88.86 (0.09) | 39.73 (0.53) | 42.75 (0.54) |
| tem 1.2 | 27.19 (0.50) | 88.82 (0.17) | 32.10 (0.42) | 46.36 (0.40) |
| tem 1.4 | 23.64 (0.40) | 87.84 (0.06) | 24.22 (0.12) | 51.23 (0.53) |
| topp 0.8 | 31.90 (0.18) | 92.15 (0.06) | 52.82 (0.67) | 35.45 (0.48) |
| topp 0.85 | 31.36 (0.29) | 92.09 (0.04) | 48.54 (1.02) | 36.68 (0.35) |
| topp 0.9 | 31.52 (0.71) | 91.96 (0.03) | 45.62 (0.55) | 38.20 (0.52) |
| topp 0.95 | 29.92 (0.28) | 91.67 (0.02) | 41.17 (0.73) | 39.95 (0.26) |
| topp 1.0 | 29.28 (0.47) | 91.04 (0.06) | 36.73 (0.86) | 41.31 (0.37) |
| beam 1 penalty 0.6 | 31.21 | 88.48 | 58.83 | 43.21 |
| beam 1 penalty 0.8 | 30.88 | 87.04 | 53.23 | 46.09 |
| beam 1 penalty 1.0 | 28.64 | 86.32 | 49.96 | 47.95 |
| beam 2 penalty 0.6 | 28.68 | 86.96 | 61.28 | 45.53 |
| beam 2 penalty 0.8 | 28.96 | 86.75 | 56.94 | 48.07 |
| beam 2 penalty 1.0 | 27.22 | 84.90 | 55.46 | 49.54 |

Table 13: SQuAD1: Standard decoding baseline results

We evaluate quality via validity and answer loss. Validity measures whether generated SMILES strings can be converted to valid molecule objects using RDKit [1]. Answer loss uses `LlaSMol-Mistral-7B` (Yu et al., 2024) to generate molecule descriptions, then calculates prediction loss against reference descriptions. Comprehensive results can be found in Table 19-21

## H.5 Forward Synthesis Molecule Generation (FS-Mol)

We additionally use a quality metric called Tanimoto Similarity (Bajusz et al., 2015) that quantifies structural similarity between generated and reference products. Comprehensive results can be found in Table 22-24

## I Computational Resources

All experiments are conducted on high-performance computing clusters. We used 1 NVIDIA A100 GPU for each open-source models we mentioned and used in the Experiment section.

---

[1]RDKit: Open-source cheminformatics. https://www.rdkit.org

| Hyperparameters | Quality Metrics | | Diversity Metrics | |
|---|---|---|---|---|
| | O. BLEU ↑ | UniEval (%) ↑ | P. BLEU ↓ | Vendi (%) ↑ |
| tem 0.6 | 33.14 (0.40) | 92.22 (0.04) | 56.94 (0.71) | 34.73 (0.33) |
| tem 0.8 | 32.88 (0.18) | 92.00 (0.11) | 45.76 (0.47) | 38.22 (0.45) |
| tem 1.0 | 29.61 (0.68) | 91.45 (0.08) | 36.19 (0.48) | 42.58 (0.27) |
| tem 1.2 | 26.19 (0.21) | 90.12 (0.14) | 27.34 (1.02) | 48.08 (0.82) |
| tem 1.4 | 22.12 (0.51) | 87.26 (0.25) | 19.38 (0.40) | 55.21 (0.57) |
| topp 0.8 | 33.03 (0.25) | 91.93 (0.05) | 46.81 (0.67) | 38.02 (0.17) |
| topp 0.85 | 31.62 (0.50) | 91.95 (0.09) | 43.42 (0.43) | 39.59 (0.15) |
| topp 0.9 | 30.86 (0.46) | 91.72 (0.06) | 39.76 (0.16) | 41.14 (0.18) |
| topp 0.95 | 29.56 (0.37) | 91.46 (0.02) | 35.86 (0.35) | 42.88 (0.39) |
| topp 1.0 | 28.26 (0.36) | 90.45 (0.08) | 32.04 (0.54) | 45.37 (0.55) |
| beam 1 penalty 0.6 | 32.39 | 89.81 | 58.64 | 43.37 |
| beam 1 penalty 0.8 | 31.47 | 88.39 | 53.36 | 46.67 |
| beam 1 penalty 1.0 | 29.72 | 87.31 | 49.57 | 48.55 |
| beam 2 penalty 0.6 | 29.89 | 89.43 | 62.46 | 45.79 |
| beam 2 penalty 0.8 | 28.86 | 88.06 | 60.29 | 48.26 |
| beam 2 penalty 1.0 | 28.26 | 86.55 | 57.34 | 49.60 |

Table 14: SQuAD 1: Soft prompt tuning results, to be compared with the standard decoding baseline

| Method | Quality Metrics | | Diversity Metrics | |
|---|---|---|---|---|
| | O. BLEU ↑ | UniEval (%) ↑ | P. BLEU ↓ | Vendi (%) ↑ |
| gpt-4o-mini | 15.90 | 92.87 | 61.52 | 33.00 |
| Temperature Sampling | 24.30 | 91.09 | 33.16 | 44.60 |
| Ours (temp) | 23.39 | 90.12 | 27.40 | 48.28 |
| Top-p Sampling | 25.36 | 91.14 | 36.76 | 42.60 |
| Ours (top-p) | 24.59 | 90.47 | 31.01 | 45.44 |
| Diverse Beam Search | 28.45 | 87.10 | 51.87 | 47.22 |
| Ours (beam) | 29.55 | 88.39 | 53.42 | 47.35 |
| Ours | 23.39 | 90.12 | 27.40 | 48.28 |
| Randomly initialized soft prompt | 24.80 | 90.55 | 29.75 | 45.80 |
| Hidden state noise injection | 24.43 | 91.06 | 33.19 | 44.21 |

Table 15: SQuAD 1: Comparison between soft prompt tuning method with other baselines

| Hyperparameters | Quality Metrics | | Diversity Metrics | |
|---|---|---|---|---|
| | O. BLEU ↑ | UniEval (%) ↑ | P. BLEU ↓ | Vendi (%) ↑ |
| tem 0.6 | 27.06 (0.41) | 91.66 (0.09) | 62.43 (0.34) | 31.49 (0.28) |
| tem 0.8 | 27.26 (0.40) | 91.42 (0.08) | 52.33 (0.32) | 34.52 (0.17) |
| tem 1.0 | 25.13 (0.30) | 90.90 (0.07) | 43.19 (0.31) | 38.21 (0.49) |
| tem 1.2 | 22.88 (0.22) | 89.89 (0.07) | 34.34 (0.07) | 42.58 (0.23) |
| tem 1.4 | 20.31 (0.27) | 87.99 (0.05) | 26.13 (0.26) | 47.70 (0.66) |
| topp 0.8 | 26.54 (0.24) | 91.40 (0.07) | 54.20 (0.37) | 33.69 (0.33) |
| topp 0.85 | 26.26 (0.54) | 91.43 (0.03) | 50.08 (0.44) | 35.27 (0.29) |
| topp 0.9 | 26.07 (0.39) | 91.18 (0.03) | 46.56 (0.84) | 36.22 (0.24) |
| topp 0.95 | 25.31 (0.49) | 90.91 (0.12) | 42.79 (0.33) | 38.02 (0.30) |
| topp 1.0 | 24.64 (0.36) | 90.00 (0.22) | 37.71 (0.39) | 40.25 (0.36) |
| beam 1 penalty 0.6 | 27.06 | 86.64 | 56.95 | 44.75 |
| beam 1 penalty 0.8 | 26.47 | 85.73 | 52.18 | 47.79 |
| beam 1 penalty 1.0 | 26.11 | 84.31 | 47.93 | 51.56 |
| beam 2 penalty 0.6 | 25.87 | 84.10 | 58.35 | 48.94 |
| beam 2 penalty 0.8 | 24.79 | 83.24 | 55.56 | 51.23 |
| beam 2 penalty 1.0 | 24.45 | 82.52 | 51.93 | 54.75 |

Table 16: SQuAD2: Standard decoding baseline results

| Hyperparameters | Quality Metrics | | Diversity Metrics | |
|---|---|---|---|---|
| | O. BLEU ↑ | UniEval (%) ↑ | P. BLEU ↓ | Vendi (%) ↑ |
| tem 0.6 | 28.38 (0.21) | 91.55 (0.01) | 59.40 (1.00) | 32.87 (0.12) |
| tem 0.8 | 27.29 (0.19) | 91.16 (0.06) | 47.49 (0.14) | 36.83 (0.43) |
| tem 1.0 | 25.29 (0.42) | 90.47 (0.07) | 36.93 (0.79) | 40.44 (0.33) |
| tem 1.2 | 22.58 (0.30) | 88.75 (0.13) | 29.02 (0.09) | 45.62 (0.65) |
| tem 1.4 | 19.41 (0.56) | 85.71 (0.22) | 20.54 (0.20) | 52.42 (0.17) |
| topp 0.8 | 27.34 (0.44) | 91.24 (0.06) | 49.01 (0.38) | 35.82 (0.29) |
| topp 0.85 | 26.65 (0.22) | 91.10 (0.02) | 44.65 (0.89) | 37.72 (0.17) |
| topp 0.9 | 26.04 (0.59) | 90.81 (0.06) | 41.61 (0.69) | 39.14 (1.02) |
| topp 0.95 | 25.92 (0.57) | 90.49 (0.11) | 37.74 (0.38) | 40.96 (0.44) |
| topp 1.0 | 23.59 (0.08) | 89.15 (0.04) | 32.44 (0.27) | 44.10 (0.44) |
| beam 1 penalty 0.6 | 27.76 | 88.19 | 56.82 | 46.40 |
| beam 1 penalty 0.8 | 26.31 | 86.88 | 50.36 | 50.26 |
| beam 1 penalty 1.0 | 26.38 | 86.48 | 47.89 | 51.94 |
| beam 2 penalty 0.6 | 26.15 | 87.74 | 59.80 | 48.86 |
| beam 2 penalty 0.8 | 25.59 | 86.34 | 57.50 | 50.88 |
| beam 2 penalty 1.0 | 23.80 | 85.25 | 56.67 | 52.77 |

Table 17: SQuAD 2: Soft prompt tuning results, to be compared with the standard decoding baseline

| Method | Quality Metrics | | Diversity Metrics | |
|---|---|---|---|---|
| | O. BLEU ↑ | UniEval (%) ↑ | P. BLEU ↓ | Vendi (%) ↑ |
| gpt-4o-mini | 17.30 | 91.74 | 65.19 | 16.84 |
| Temperature Sampling | 24.12 | 89.88 | 39.51 | 18.27 |
| Ours (temp) | 23.53 | 88.87 | 33.87 | 20.00 |
| Top-p Sampling | 25.13 | 89.97 | 43.18 | 17.53 |
| Ours (top-p) | 24.30 | 89.28 | 38.12 | 18.45 |
| Diverse Beam Search | 18.66 | 85.46 | 51.86 | 49.00 |
| Ours (beam) | 19.73 | 86.91 | 50.78 | 50.79 |
| Ours | 24.12 | 89.88 | 39.51 | 18.27 |
| Randomly initialized soft prompt | 22.93 | 87.29 | 36.55 | 25.97 |
| Hidden state noise injection | 24.05 | 89.94 | 39.76 | 18.24 |

Table 18: SQuAD 2: Comparison between soft prompt tuning method with other baselines

| Hyperparameters | Quality Metrics | | Diversity Metrics | |
|---|---|---|---|---|
| | Validity (%) ↑ | Ans. Loss ↓ | NCircle (h=0.75) (%) ↑ | Vendi (%) ↑ |
| tem 0.6 | 99.79 (0.05) | 2.28 (0.00) | 16.42 (0.47) | 4.13 (0.02) |
| tem 0.8 | 99.52 (0.13) | 2.27 (0.01) | 27.21 (0.22) | 5.60 (0.08) |
| tem 1.0 | 98.94 (0.07) | 2.27 (0.00) | 43.59 (0.46) | 7.12 (0.06) |
| tem 1.2 | 97.97 (0.14) | 2.27 (0.00) | 61.74 (0.79) | 8.71 (0.10) |
| tem 1.4 | 95.09 (0.37) | 2.28 (0.01) | 76.93 (0.42) | 10.25 (0.09) |
| topp 0.8 | 99.61 (0.07) | 2.28 (0.00) | 23.88 (0.27) | 4.71 (0.03) |
| topp 0.85 | 99.34 (0.09) | 2.27 (0.00) | 28.24 (0.17) | 5.19 (0.04) |
| topp 0.9 | 99.27 (0.06) | 2.27 (0.00) | 34.62 (0.12) | 5.84 (0.04) |
| topp 0.95 | 99.01 (0.06) | 2.26 (0.00) | 43.68 (0.10) | 6.45 (0.03) |
| topp 1.0 | 97.54 (0.15) | 2.28 (0.00) | 54.63 (0.70) | 7.17 (0.01) |
| beam 1 penalty 0.6 | 94.82 | 2.37 | 63.48 | 14.49 |
| beam 1 penalty 0.8 | 92.60 | 2.41 | 69.42 | 16.52 |
| beam 1 penalty 1.0 | 89.98 | 2.44 | 71.22 | 17.94 |
| beam 2 penalty 0.6 | 94.26 | 2.43 | 50.40 | 12.57 |
| beam 2 penalty 0.8 | 92.06 | 2.45 | 52.86 | 13.94 |
| beam 2 penalty 1.0 | 88.88 | 2.48 | 52.54 | 15.32 |

Table 19: Desc-Mol: Standard decoding baseline results

| Hyperparameters | Quality Metrics | | Diversity Metrics | |
|---|---|---|---|---|
| | Validity (%) ↑ | Ans. Loss ↓ | NCircle (h=0.75) (%) ↑ | Vendi (%) ↑ |
| tem 0.6 | 99.71 (0.01) | 2.29 (0.00) | 15.65 (0.03) | 4.01 (0.01) |
| tem 0.8 | 99.44 (0.00) | 2.27 (0.00) | 27.27 (0.01) | 5.18 (0.00) |
| tem 1.0 | 98.93 (0.01) | 2.27 (0.00) | 43.77 (0.03) | 6.38 (0.00) |
| tem 1.2 | 97.72 (0.02) | 2.27 (0.00) | 63.34 (0.06) | 7.44 (0.00) |
| tem 1.4 | 94.59 (0.02) | 2.29 (0.00) | 78.21 (0.01) | 8.07 (0.01) |
| topp 0.8 | 99.66 (0.00) | 2.28 (0.00) | 24.25 (0.01) | 4.60 (0.01) |
| topp 0.85 | 99.55 (0.01) | 2.27 (0.00) | 28.35 (0.02) | 5.16 (0.00) |
| topp 0.9 | 99.13 (0.02) | 2.26 (0.00) | 35.84 (0.04) | 5.84 (0.01) |
| topp 0.95 | 98.94 (0.00) | 2.27 (0.00) | 43.79 (0.02) | 6.38 (0.00) |
| topp 1.0 | 97.49 (0.04) | 2.29 (0.00) | 56.42 (0.13) | 7.04 (0.00) |
| beam 1 penalty 0.6 | 94.18 | 2.39 | 64.74 | 14.40 |
| beam 1 penalty 0.8 | 92.46 | 2.42 | 70.56 | 16.28 |
| beam 1 penalty 1.0 | 90.72 | 2.45 | 73.02 | 18.08 |
| beam 2 penalty 0.6 | 93.72 | 2.44 | 50.84 | 12.35 |
| beam 2 penalty 0.8 | 91.46 | 2.45 | 52.14 | 13.74 |
| beam 2 penalty 1.0 | 89.40 | 2.48 | 52.18 | 15.04 |

Table 20: Desc-Mol: Soft prompt tuning results, to be compared with the standard decoding baseline

| Method | Quality Metrics | | Diversity Metrics | |
|---|---|---|---|---|
| | Validity (%) ↑ | Ans. Loss ↓ | NCircle (h=0.75) (%) ↑ | Vendi (%) ↑ |
| gpt-4o-mini | 72.83 | 1.97 | 46.40 | 30.99 |
| Temperature Sampling | 97.99 | 2.36 | 63.16 | 7.20 |
| Ours (temp) | 97.67 | 2.36 | 64.68 | 7.20 |
| Top-p Sampling | 97.57 | 2.37 | 56.35 | 7.10 |
| Ours (top-p) | 97.39 | 2.37 | 56.73 | 7.00 |
| Diverse Beam Search | 98.14 | 2.46 | 70.24 | 18.45 |
| Ours (beam) | 98.29 | 2.47 | 71.30 | 16.22 |
| Ours | 97.67 | 2.36 | 64.68 | 7.20 |
| Randomly initialized soft prompt | 97.33 | 2.36 | 63.67 | 7.20 |
| Hidden state noise injection | 96.03 | 2.36 | 67.46 | 8.60 |

Table 21: Desc-Mol: Comparison between soft prompt tuning method with other baselines

| Hyperparameters | Quality Metrics | | Diversity Metrics | |
|---|---|---|---|---|
| | Tanimoto Sim. ↑ | Confidence ↑ | NCircle (h=0.75) (%) ↑ | Vendi (%) ↑ |
| tem 0.6 | 0.49 (0.00) | 0.42 (0.00) | 5.95 (0.03) | 3.19 (0.01) |
| tem 0.8 | 0.46 (0.00) | 0.40 (0.00) | 8.65 (0.10) | 3.80 (0.01) |
| tem 1.0 | 0.42 (0.00) | 0.38 (0.00) | 12.51 (0.13) | 4.55 (0.02) |
| tem 1.2 | 0.37 (0.00) | 0.35 (0.00) | 18.91 (0.07) | 5.37 (0.02) |
| tem 1.4 | 0.32 (0.00) | 0.31 (0.00) | 28.56 (0.28) | 6.21 (0.01) |
| topp 0.8 | 0.47 (0.00) | 0.41 (0.00) | 7.06 (0.08) | 3.56 (0.01) |
| topp 0.85 | 0.45 (0.00) | 0.40 (0.00) | 8.60 (0.16) | 3.85 (0.00) |
| topp 0.9 | 0.44 (0.00) | 0.39 (0.00) | 10.17 (0.07) | 4.17 (0.00) |
| topp 0.95 | 0.42 (0.00) | 0.38 (0.00) | 12.63 (0.03) | 4.56 (0.01) |
| topp 1.0 | 0.38 (0.00) | 0.36 (0.00) | 17.86 (0.10) | 5.04 (0.03) |
| beam 1 penalty 0.6 | 0.21 | 0.29 | 43.85 | 13.54 |
| beam 1 penalty 0.8 | 0.18 | 0.28 | 51.97 | 15.58 |
| beam 1 penalty 1.0 | 0.16 | 0.28 | 57.58 | 17.26 |
| beam 2 penalty 0.6 | 0.20 | 0.29 | 35.93 | 12.20 |
| beam 2 penalty 0.8 | 0.16 | 0.30 | 42.05 | 13.60 |
| beam 2 penalty 1.0 | 0.15 | 0.31 | 45.27 | 15.08 |

Table 22: FS-Mol: Standard decoding baseline results

| Hyperparameters | Quality Metrics | | Diversity Metrics | |
|---|---|---|---|---|
| | Tanimoto Sim. ↑ | Confidence ↑ | NCircle (h=0.75) (%) ↑ | Vendi (%) ↑ |
| tem 0.6 | 0.47 (0.01) | 0.44 (0.01) | 6.86 (0.17) | 3.43 (0.05) |
| tem 0.8 | 0.43 (0.00) | 0.42 (0.01) | 9.80 (0.27) | 4.14 (0.10) |
| tem 1.0 | 0.40 (0.01) | 0.40 (0.01) | 14.76 (0.17) | 4.91 (0.01) |
| tem 1.2 | 0.35 (0.01) | 0.37 (0.01) | 21.81 (0.04) | 5.78 (0.01) |
| tem 1.4 | 0.30 (0.00) | 0.33 (0.01) | 32.30 (0.13) | 6.63 (0.03) |
| topp 0.8 | 0.44 (0.00) | 0.42 (0.02) | 8.54 (0.27) | 3.96 (0.08) |
| topp 0.85 | 0.43 (0.00) | 0.42 (0.02) | 9.93 (0.36) | 4.23 (0.05) |
| topp 0.9 | 0.41 (0.01) | 0.40 (0.01) | 11.91 (0.15) | 4.56 (0.06) |
| topp 0.95 | 0.40 (0.01) | 0.40 (0.02) | 14.72 (0.17) | 4.92 (0.02) |
| topp 1.0 | 0.36 (0.01) | 0.38 (0.02) | 20.46 (0.03) | 5.49 (0.01) |
| beam 1 penalty 0.6 | 0.20 | 0.31 | 45.82 | 13.98 |
| beam 1 penalty 0.8 | 0.17 | 0.31 | 52.78 | 16.09 |
| beam 1 penalty 1.0 | 0.15 | 0.31 | 57.07 | 17.75 |
| beam 2 penalty 0.6 | 0.18 | 0.33 | 37.29 | 13.02 |
| beam 2 penalty 0.8 | 0.15 | 0.34 | 42.67 | 14.46 |
| beam 2 penalty 1.0 | 0.14 | 0.34 | 45.74 | 15.85 |

Table 23: FS-Mol: Soft prompt tuning results, to be compared with the standard decoding baseline

| Method | Quality Metrics | | Diversity Metrics | |
|---|---|---|---|---|
| | Tanimoto Sim. ↑ | Confidence ↑ | NCircle (h=0.75) (%) ↑ | Vendi (%) ↑ |
| gpt-4o-mini | 0.10 | 0.08 | 1.81 | 2.41 |
| Temperature Sampling | 0.38 | 0.36 | 19.00 | 5.43 |
| Ours (temp) | 0.35 | 0.36 | 21.42 | 5.81 |
| Top-p Sampling | 0.39 | 0.37 | 17.82 | 5.10 |
| Ours (top-p) | 0.36 | 0.37 | 19.99 | 5.50 |
| Diverse Beam Search | 0.18 | 0.29 | 52.41 | 15.46 |
| Ours (beam) | 0.17 | 0.31 | 52.94 | 15.92 |
| Ours | 0.35 | 0.36 | 21.42 | 5.81 |
| Randomly initialized soft prompt | 0.34 | 0.34 | 22.13 | 5.73 |
| Hidden state noise injection | 0.31 | 0.33 | 26.01 | 7.51 |

Table 24: FS-Mol: Comparison between soft prompt tuning method with other baselines.

