# OpenReview forum: "Diverse Text Generation through Soft Prompt Tuning"
_ICLR.cc/2026/Conference — Submitted to ICLR 2026_

### Official Review · Reviewer_Pixg · 2025-10-24

**Soundness:** 2
**Presentation:** 2
**Contribution:** 2
**Rating:** 4
**Confidence:** 4

**Summary:**

The paper introduces a diverse generation technique taking advantage of soft prompt vectors. It breaks down an input into context prompt and generation prompt. Soft prompts are inserted between these two parts to encourage the model to produce diverse outputs. Tuning objective consists hidden state diversity and coherence, calculated as L2 distances.

Experiments on combinatorial generation, question generation, and molecular generation indicate that their tuned generation method achieves better trade-off between quality and diversity than do classic sampling methods.

**Strengths:**

- The proposed framework is parameter-efficient and easy to train.
- The training objective is intuitive, as it promotes diversity and balances coherence by tracking the hidden states.
- Experiment results show Pareto improvement.

**Weaknesses:**

- The baselines are weak. The paper compares against naive sampling methods such as temperature and nucleus sampling. No advanced methods, like generating based on diverse natural language prefixes, or any method mentioned from the related work section, are included as baselines.
- It is unclear what data is used to tune the soft prompts for each experiment.
- The paper criticizes RL methods for not transferring, but the tuned soft prompts are also task-specific, as admitted by the authors.
- Lester et al. [1] demonstrate the zero-shot transfer capability of soft prompts, but that is not demonstrated here. This suggests these diversity-targeted soft prompts may not be generalizable.
- More analysis on the soft prompt vectors themselves beyond just the L2 distances would be helpful for understanding. For example, what are the directions of these vectors? what does inverting a soft prompt vector produce?

[1] The Power of Scale for Parameter-Efficient Prompt Tuning (Lester et al., 2021)

**Questions:**

- Is “diversity as the distance of hidden states” a validated proxy for semantic diversity? The introduced method relies entirely on this assumption, but it is not well-justified.
- Why was L2 distance chosen for the diversity loss? The L2 norm is not agnostic to the magnitude of the hidden state vectors. Why was this used instead of a standard magnitude-invariant metric like cosine similarity?
- Why is SGD used for optimization? Is an adaptive optimizer like Adam considered, which might simply be better?
-  Can a set of general-purpose diversity prompts be learned?

---

> ### Author Response · Authors · 2025-12-03
> **Official Response to Reviewer Pixg**
>
> Hi Reviewer Pixg,
>
> Thank you so much for your insightful and detailed comments! Here are our responses to your concerns and questions raised:
>
> > **Weakness 1:** The baselines are weak. The paper compares against naive sampling methods such as temperature and nucleus sampling. No advanced methods, like generating based on diverse natural language prefixes, or any method mentioned from the related work section, are included as baselines.
> >
>
> As requested, we tried the diverse natural language prefixes baselines as you suggested. Concretely, for each task we design a set of short natural language prefixes that explicitly ask the model to produce diverse outputs, and we prepend one such prefix to the generation prompt for each desired sample. It is important to note that these prefixes need to respect different diversifying strategies.
>
> For example, for combinatorial tasks, we address diverse selection strategies (balanced vs exploratory). For question generation, we address diverse question styles (investigative vs conversational tone). For Molecule generation, we address diverse structural hypotheses. The number of the task-specific prefixes equals the number of desired generations so that each generation is steered by a distinct prefix to maximize diversity.
>
> Below we demonstrate the results compared to our methods, we can observe that diverse prefixes elicit more diverse responses as expected, but this comes at a great cost of response quality. Moreover, in realistic applications, these prefixes must be manually designed and tuned for each task to simultaneously respect both task-specific nuanced quality and diversity aim, which significantly limits their generality compared to our learned soft-prompt approach.
>
> **(1) Combinatorial Task**
>
> | **Combinatorial Task** | **20% Acpt (↑)** | **50% Acpt (↑)** | **MRSE (↓)** | **Uniqueness (%) (↑)** | **Vendi (%) (↑)** |
> | --- | --- | --- | --- | --- | --- |
> | Diverse Prefix | 23.05 | 53.6 | 0.85 | **99.66** | **14.86** |
> | Ours | **28.06** | **67.42** | **0.44** | 98.53 |  17.13 |
>
> **(2) Question Generation Task**
>
> | **QG1 Task** | **O.BLEU (↑)** | **UniEval (%) (↑)** | **P.BLEU (↓)** | **Vendi (%) (↑)** |
> | --- | --- | --- | --- | --- |
> | Diverse Prefix | 18.16 | 77.33 | **14.1** | **83.31** |
> | Ours | **23.39** | **90.12** | 27.4 | 48.28 |
>
> **(3) Molecule Generation Task**
>
> | **Molecule Task** | **Tanimoto Sim. (↑)** | **Confidence (↑)** | **NCircle (h=0.75) (%) (↑)** | **Vendi (%) (↑)** |
> | --- | --- | --- | --- | --- |
> | Diverse Prefix | 0.29 | 0.3 | **31.01** | **6.94** |
> | Ours | **0.35** | **0.36** | 21.42 | 5.81 |
>
> > **Weakness 2:** It is unclear what data is used to tune the soft prompts for each experiment.
> >
>
> We would like to clarify that our methods **do not rely on any external or conventional training data for soft prompt tuning**. The soft prompts are optimized at test time for each input instance. In particular, if we aim to generate *n* diverse responses over one prompt, we would initialize *n* different soft prompts and optimize them using our objective based on the hidden states they elicit, as well as a hidden state from an input without a soft prompt. This training process does not require external labeled data and the hidden states are retrieved solely from the test instance itself. We will clarify this more explicitly in the revised manuscript by stating that soft prompts are tuned per input at test time, without any separate dataset.

---

> > ### Author Response · Authors · 2025-12-03
> > **Official Response to Reviewer Pixg - continued**
> >
> > > **Weakness 3:** The paper criticizes RL methods for not transferring, but the tuned soft prompts are also task-specific, as admitted by the authors.
> > >
> >
> > A major reason why we mentioned the RL-based methods is that the reward design and training pipeline in RL does not transfer easily across tasks, whereas our training objective and pipeline are more easily adaptive.
> >
> > Recent RL-based diversification methods typically require task-specific reward functions that encode diversity and task-specific quality indicators. As the complicated rewards are being integrated, the training process could face training instability challenges. RL training process also faces challenges like higher sensitivity about more hyperparameters, higher computational costs, which is even more challenging when the rewards are verifiable but expensive, such as calling external tools in the molecule generation task.
> >
> > In contrast, our methods use a single, task-agnostic objective that works purely on hidden states. We agree that our current methods do not yield a universal diversifying soft prompt set yet, but the mechanism for obtaining is simple and reusable across tasks, unlike RL pipelines that require major redesign and tuning.
> >
> > > **Weakness 5:** More analysis on the soft prompt vectors themselves beyond just the L2 distances would be helpful for understanding. For example, what are the directions of these vectors? what does inverting a soft prompt vector produce?
> > >
> >
> > We appreciate the suggestion to further analyze the learned soft prompts and agree that this would be an interesting topic to explore, though it is somewhat outside the main contribution of our work. Since soft prompts are high-dimensional $(\mathbb{R}^{n \times d}$, where $n$ is the soft prompt length and $d$ is the model embedding dimension), their “directions” may not be easily interpretable and are hard to compare. It would even be harder if we want to interpret the directions across tasks because we empirically use different soft prompt lengths across different tasks. Intuitively, inverting a trained soft prompt would push the hidden states away from the regions that jointly improve diversity and quality, but it is difficult to measure the collective effect since we have multiple distinct trained soft prompts. Nonetheless, systematically characterizing soft prompts behaviors through interpolation experiments or probing prompts in a shared low-dimensional basis would be an interesting interpretability direction for future work.
> >
> > > **Weakness 4 & Question 4:** Limited generalizability of soft prompts & Can a set of general-purpose diversity prompts be learned?
> > >
> >
> > Please refer to our response to Reviewer UaGR regarding Weakness 2.
> >
> >
> >
> > > **Question 1:** Is “diversity as the distance of hidden states” a validated proxy for semantic diversity?
> > >
> >
> > Please refer to our response to Reviewer vCio.
> >
> > > **Question 2:** Why was L2 distance chosen for the diversity loss?
> > >
> >
> > We chose an L2-based distance for the diversity loss mainly for simplicity and stability. Because we are always comparing hidden states from the same layer of the same transformer, their norms are already fairly well controlled, so L2 distance is in practice highly correlated with cosine similarity while yielding reliable performance.
> >
> > > **Question 3:** Why is SGD used for optimization?
> > >
> >
> > We use SGD because plain SGD with a tuned learning rate already converges reliably and keeps the method simple and reproducible. In preliminary experiments we did not observe a consistent improvement from Adam once SGD was reasonably tuned, we therefore kept SGD as the default in this work.

---

### Official Review · Reviewer_W9rH · 2025-10-26

**Soundness:** 3
**Presentation:** 2
**Contribution:** 3
**Rating:** 4
**Confidence:** 4

**Summary:**

This paper proposes a new method to make language models produce a wider variety of high-quality responses from a single input prompt. Via an informed initialization scheme, followed by a fine-tuning stage, a set of soft prompts are trained to yield diverse hidden states from one another. This is achieved by maximizing the distance in the  later hidden states of the sequence, whilst keeping the distance in earlier hidden states close.

The method was tested on three distinct tasks: a novel combinatorial "target-sum" task, question generation, and molecular design. The authors demonstrate good results across all tasks, with their approach providing a preferred quality-diversity trade-off. In particular, the approach showed particular strength in complex tasks with large exploration spaces.

**Strengths:**

- The simplicity, and therefore the applicability, of the method is strong and should make for easy implementation.
- The results (although not always easily digested) are promising.
- For the most part the paper is well written, and it paints a clear story. (See Concern 5 for suggested improvements)

**Weaknesses:**

**Concern 1:**

The paper currently does not consider sequentially generating diverse output with a single model. If the goal is to generate a set of diverse outputs to a given prompt, it would be possible to have an LLM see its previous outputs in its context, rather than the parallel approach provided by this paper. This would be extra appealing considering the large context windows of contemporary LLMs, and that this approach would require no fine-tuning or soft prompts.

This sequential approach of course has potential downsides, and I’m not sure it would yield the same diversity. But since a strong part of this paper is about the method being lightweight, it would significantly strengthen the paper if you could demonstrate how your more complicated method outperforms the trivial sequential one.

**Concern 2:**

In line with concern 1, it would greatly benefit the paper if you could demonstrate the usefulness of this approach in scenarios where diversity is crucial, and not only desired. I would suggest perhaps trying this approach in an RL setting with LLMs. There has been some recent work showing how the LLM entropy has an undesirable collapse during RL training, which negatively impacts exploration.
https://arxiv.org/abs/2505.22617


Such a scenario would also arguably be harder to solve using the straight-forward sequential approach in concern 1. And might therefore be a strong point for your paper (given that Concern 3 is not overwhelming)


**Concern 3:**

There is no explicit information regarding the computational overhead for this method. Since the authors argue for their method being “lightweight”, providing empirical evidence corroborating this claim would strengthen the paper.  In particular, this would be interesting to see in regards to concern 1.

**Concern 4:**

Figure 3 is quite messy at the moment. Perhaps moving the Beam-Search result into a separate figure?

Also, what are the crown icons for?!?! It does not provide any value in my opinion, and only makes an already busy figure more cluttered.

**Concern 5: (minor)**

Some parts of the paper could benefit from having an extra iteration of the text.

Line 32: Language Models (LLM), should be either Large Language Models (LLM), or change the abbreviation to (LM).

Line 155: You start by explaining that the intuition is two-fold, but what follows immediately is more concerned with “what” you have done. Not why you do it.

Line 291-296: This paragraph is a bit unclear for a first time reader. It introduces the task and scenarios, but it’s not entirely clear what these scenarios are. I suggest adding a figure that displays one (or several) qualitative examples of the scenarios you’re considering.

**Questions:**

Considering that this approach requires one to train a set of prompts for each specific task. Do you think it would be possible to generate a set of "general" diversity prompts. Meaning that they could be used for a wide range of tasks.

Ideally, one could imagine that the task of these prompts would be to condition the model in some sense, so that prompt A yields a distinct output to prompt B etc... Whilst the quality remain intact. (as your approach does now)

---

> ### Author Response · Authors · 2025-12-03
> **Official Response to Reviewer W9rH**
>
> Hi Reviewer W9rH,
>
> Thank you so much for your thoughtful and detailed comments, and they really helped us re-think the main focus of our method. Please find our responses to your concerns and questions below.
>
> > **Concern 1: Long context sequential generation baseline.**
> >
>
> We agree that this is a valuable baseline in principle. However, implementing it efficiently is non-trivial in our current HuggingFace-based generation setup, because providing historical responses back to the context forces us to reduce the effective batch size. We can no longer generate all desired samples for one instance in a single batched call. Across tasks, we target 20, 50, and 100 generations per instance for the combinatorial, question generation, and molecule generation tasks, respectively. Under our original batched configuration, the amortized per-instance time is about **9s** (combinatorial), **2s** (question generation), and **6s** (molecule generation). With full history and batch size = 1, the per-instance time grows to **2m53s**, **40s**, and **6m**, respectively; even with larger but still constrained batch sizes of 10 / 25 / 50, the per-instance times are still around **26s**, **6s**, and **16s**.
>
> Below we report the results for the larger batch size setting due to feasibility. Overall, the long-context sequential baseline is competitive but does not dominate our method. On the combinatorial task, it achieves a higher 20% acceptance rate and slightly higher Vendi, while our approach yields a better 50% acceptance and noticeably lower MRSE, with essentially identical uniqueness. For question generation, the two methods have very similar UniEval scores, while the long context baselines outperform in the BLEU score metrics. For molecule generation, our method produces candidates of higher quality with higher Tanimoto similarity and reaction confidence, whereas the long-context baseline attains higher NCircle and Vendi at the cost of quality.
>
> **(1) Combinatorial Task**
>
> | **Combinatorial Task** | **20% Acpt (↑)** | **50% Acpt (↑)** | **MRSE (↓)** | **Uniqueness (%) (↑)** | **Vendi (%) (↑)** |
> | --- | --- | --- | --- | --- | --- |
> | Long Context | 34.97 | 66.48 | 0.54 | 98.6 | 18.44 |
> | Ours | 28.06 | 67.42 | 0.44 | 98.53 | 17.13 |
>
> **(2) Question Generation Task**
>
> | **QG1 Task** | **O.BLEU (↑)** | **UniEval (%) (↑)** | **P.BLEU (↓)** | **Vendi (%) (↑)** |
> | --- | --- | --- | --- | --- |
> | Long Context | 25.44 | 90.56 | 25.66 | 52.96 |
> | Ours | 23.39 | 90.12 | 27.4 | 48.28 |
>
> **(3) Molecule Generation Task**
>
> | **Molecule Task** | **Tanimoto Sim. (↑)** | **Confidence (↑)** | **NCircle (h=0.75) (%) (↑)** | **Vendi (%) (↑)** |
> | --- | --- | --- | --- | --- |
> | Long Context | 0.32 | 0.3 | 26.98 | 6.6 |
> | Ours | 0.35 | 0.36 | 21.42 | 5.81 |

---

> > ### Author Response · Authors · 2025-12-03
> > **Official Response to Reviewer W9rH - continued**
> >
> > > **Concern 2: Application to RL-based LM post-training scenarios, where soft prompts are crucial rather than merely desirable.**
> > >
> >
> > We really appreciate the raised idea of including an application scenario where diversity is crucial. Following the suggestion, we post-train and evaluate Qwen2-0.5b-Instruct on the combinatorial task we designed, using GRPO with a rollout size of 32. We shape the rewards to [0,1], where a higher reward indicates a higher quality response, whose selected subset sum is close to the target value. In particular, the reward is computed by first taking a base score that starts at 1 when the selected subset sum exactly matches the target value and decreases linearly as the absolute difference between the selected sum and the target increases, normalized by the magnitude of the target. We then clip this base score at 0 so the reward never becomes negative. Finally, we multiply the score by invalid_ratio, which reflects the proportion of invalid item IDs in the selection, so selections with more invalid items receive lower rewards.
> >
> > To investigate the effect of soft prompts on RL training, we first train a set of soft prompts that are later prepended to the input during the rollout phase to elicit diverse behaviors from the LM under the combinatorial task setting. To avoid overfitting the policy to a distribution where a soft prompt is always present, we linearly reduce the probability of prepending a soft prompt when rolling out from 1.0 to 0.0 over the total course of RL training, so that the final model can be used without any soft prompt at inference time. We also conduct evaluations with no soft prompt added.
> >
> > We report best@8 on the validation set. The table below compares our soft prompt-guided exploration to a GRPO baseline without soft prompts:
> >
> > |  | **Step 0** | **Step 50**  | **Step 100** | **Step 150** | **Step 200** | **Step 250** | **Step 300** |
> > | --- | --- | --- | --- | --- | --- | --- | --- |
> > | **Soft prompt-guided exploration** | 0.64 | 0.75 | 0.86 | 0.84 | 0.88 | 0.83 | 0.85 |
> > | **Baseline** | 0.65 | 0.69 | 0.75 | 0.78 | 0.80 | 0.85 | 0.85 |
> >
> > We can observe that **soft prompt–guided exploration substantially improves sample efficiency** where it reaches a best@8 of 0.86 at step 100, whereas the baseline only achieves comparable performance (0.85) around step 250. This illustrates that explicitly diversifying rollouts via soft prompts can accelerate RL fine-tuning in a setting where exploration is crucial. At the same time, we have not yet designed a fully co-trained mechanism that learns adaptive soft prompts jointly with the RL policy throughout training. Developing such a mechanism as well as designing an appropriate process-level reward signal that indicates which directions of exploration are most beneficial at a given training stage is a promising direction for future work, and we plan to explore it beyond the scope of this submission.
> >
> > > **Concern 3: Unclear computational overhead**
> > >
> >
> > Please refer to our response to Reviewer UaGR regarding Weakness 3.
> >
> > > **Concern 4 & 5: Result visualization & Manuscript edition**
> > >
> >
> > We agree with the reviewer’s suggestions regarding Figure 3 and several clarity issues in the manuscript. We will separate the beam search results from the main figure and remove unnecessary visual indicators. We will also standardize our terminology and use “LMs” consistently throughout the paper. For the two-fold intuition, we will clarify that (1) the soft prompts themselves are constructed to be diverse, and (2) the effects they induce in the model’s behavior are likewise diverse. Although we currently provide a more detailed explanation of the dataset creation process and include a qualitative example in Appendix G, we will provide more examples in the main text to improve readability.
> >
> > > **Question: Generalizability. Can a set of "general" diversity prompts be learned for a wide range of tasks?**
> > >
> >
> > Please refer to our response to Reviewer UaGR regarding Weakness 2.

---

### Official Review · Reviewer_tcyn · 2025-11-03

**Soundness:** 3
**Presentation:** 3
**Contribution:** 1
**Rating:** 4
**Confidence:** 3

**Summary:**

The paper introduces a lightweight, task-agnostic framework for diverse text generation in LLMs via soft prompt tuning. It optimizes multiple diversely initialized continuous soft prompts (using scrambled Sobol sequences), inserted between context and generation prompts, to maximize differences in final-token hidden states for diversity while minimizing early-state deviations for task coherence. This induces controlled shifts in output distributions without fine-tuning or domain-specific rewards. The method is tested on a novel combinatorial dataset (50 scenarios with item lists summing to targets), question generation (SQuAD splits), and molecular design (forward synthesis on FS-Mol, description-guided on Desc-Mol). Results show consistent Pareto improvements over baselines (temperature/top-p sampling, diverse beam search, GPT-4o-mini). Overall, it enables effective exploration in large solution spaces while adhering to constraints.

Strength:
The proposed method is sound and the empirical results are promising.

Weakness:
The novelty of the paper is limited.

**Strengths:**

The proposed method is sound and the empirical results are promising.

**Weaknesses:**

The novelty of the paper is limited. There are extensive previous work that contributed to this topic [1, 2, 3, ...]. There are not too much technical differences from the previous work. Therefore, I believe the technical contribution of this paper does not qualify for a top-tier venue like ICLR.

[1] The Power of Scale for Parameter-Efficient Prompt Tuning. Brian Lester, Rami Al-Rfou, Noah Constant. EMNLP 2021 \
[2] Selective Prompting Tuning for Personalized Conversations with LLMs. Qiushi Huang, Xubo Liu, Tom Ko, Bo Wu, Wenwu Wang, Yu Zhang, Lilian Tang. ACL 2024 findings \
[3] Controlled Text Generation using T5 based Encoder-Decoder Soft Prompt Tuning and Analysis of the Utility of Generated Text in AI. Damith Chamalke Senadeera, Julia Ive. CoRR 2022

**Questions:**

N/A

---

> ### Author Response · Authors · 2025-12-03
> **Official Response to Reviewer tcyn**
>
> Hi Reviewer tcyn,
>
> Thank you so much for your comment! Here is our response clarifying the novelty of our work:
>
> [1] This literature is the first to introduce soft prompts for parameter-efficient adaptation of frozen LMs. Although our work indeed builds upon this, we have a totally different focus and objective. Instead of improving single-output task performance, we focus on a diversity-driven task with multiple output generations. In order to achieve that, we designed novel hidden states based diversification training objectives and datasets which are not included in this literature.
>
> [2] With a totally different problem and task objective, we focus on diversification while this literature focuses on personalization. The implementations also differ intrinsically, where we do not introduce a retriever or create a large soft prompt pool selection, but focus more on the representational hidden state level of generation controls.
>
> [3] This work studies soft prompts in a T5 encoder–decoder architecture, where we focus on decoder-only causal LMs, where soft prompts act as initial conditions that shape the entire autoregressive trajectory.
>
> In summary, while we clearly build on the idea of soft prompts, we contribute a new diversity-driven hidden states level objective and a simple task-agnostic framework for multi-output diverse generation with frozen autoregressive LMs, and a new combinatorial dataset, which is shown to be challenging for LMs.

---

### Official Review · Reviewer_vCio · 2025-11-04

**Soundness:** 3
**Presentation:** 3
**Contribution:** 3
**Rating:** 4
**Confidence:** 3

**Summary:**

The paper focuses on enhancing the diversity of text generation in language models. The proposed method introduces a set of learnable prepended embeddings designed to promote diversity by maximizing the differences in the final hidden states while maintaining similarity in the earlier hidden representations. The approach is evaluated across several tasks, including combinatorial reasoning, question answering, and text generation, demonstrating modest but consistent improvements in output diversity.

**Strengths:**

The paper introduces a simple and intuitive method for improving text generation diversity in language models. Its originality lies in proposing a task-agnostic approach based on learnable prepended embeddings that promote diversity without requiring model modification or task-specific tuning. This contrasts with reinforcement learning (RL)-based methods, which often rely on carefully designed objective functions, making the proposed method broadly applicable and easy to integrate with existing models. The paper is clearly written, well organized, and supported by experimental results.

**Weaknesses:**

The paper’s technical novelty is somewhat limited. While the idea of learning prepended soft prompt embeddings is intuitive and practical, it lacks clear theoretical motivation. The link between diversity in the final hidden representations and diversity in generated text is not well established, especially given that differences in embedding space may not translate directly to token-level diversity due to the SoftMax approximation.

Additionally, the mechanism by which diverse prepended prompts lead to more diverse text outputs remains unclear. A deeper theoretical explanation or empirical analysis, such as examining intermediate representations or quantifying output diversity, would strengthen the paper’s foundation and make the proposed method more convincing.

**Questions:**

In Algorithm 1, the objective function focuses on maximizing the similarity between the embeddings of the last m hidden states of the soft prompts and those of the generation prompt. Could the authors clarify whether this objective is sufficient to ensure diversity among the soft prompt embeddings themselves?

---

> ### Author Response · Authors · 2025-12-03
> **Official Response to Reviewer vCio**
>
> Hi Reviewer vCio,
>
> Thank you so much for your insightful comment! Please find our response to your concern and question below:
>
> We appreciate the reviewer’s concerns and the opportunity to clarify this point. In short, we assume that changes in hidden states transfer to changes in token distributions, not that diverse hidden states are guaranteed to yield semantically diverse outputs. The transferability of useful diversity is demonstrated empirically by consistent gains on task-specific diversity metrics across our three tasks. Moreover, we do not attempt to formally prove a direct link between representational diversity and semantic diversity because (1) any such proof would be heavily architecture-dependent, and (2) diverse generation tasks involve richer forms of diversity than semantics alone.
>
> Due to the structure of autoregressive LLMs, the model computes the probability of each next token via a SoftMax over a linear transformation of the final-layer hidden state at the current position. This implies that changes in the hidden states of the final *m* tokens directly translate into changes in the next-token distribution. Our core assumption is therefore that **differences in hidden states lead to differences in token distributions**. However, we are not claiming a theoretically characterized equivalence that “diverse hidden states always imply semantically diverse outputs.” The subtle difference is that we assume changes in hidden states are transferable to changes in token distributions, but we do not assume transferable diversity, that every such change corresponds to a specific notion of semantic diversity.
>
> We also did not try to prove that representational diversity must yield output diversity, as this would require committing to a particular, uniform notion of diversity that may not be meaningful across tasks. Diversity metrics can vary a lot by tasks, as seen in the evaluation metrics in our three tasks. Intuitively, one would hope that semantic diversity would result in diverse outputs but this might not hold. Outputs might share similar semantics but are diverse in structural properties, such as SMILES strings in molecule tasks. Moreover, such proof can be highly architecture dependent, and a modest change in the architecture module can lead to substantial changes.
>
> With that being said, even though we generally frame our goal as generating “diverse outputs,” we view this as **task-specific diversity strategies rather than just semantic diversity.** What we do **show empirically** is that soft prompts trained to induce diverse hidden states produce outputs that are diverse in their task-specific senses. In other words, the changes that soft prompts elicit in the token distributions encode task-specific diversification strategies, and our experiments demonstrate that this representational diversity leads to useful diversity under the appropriate metrics for each task.
>
> Regarding your question, our objective in Algorithm 1 is not only a similarity objective, but it also encodes both into **control** and **diversity** terms. Inside the diversity term, we do have a specific term d_{\text{batch}} that explicitly encourages diversity among prompts. d_{\text{batch}} is the average pairwise distance between the last m hidden states across soft prompts in the batch, which we maximize in order to push different soft prompts away from each other.

---

### Official Review · Reviewer_UaGR · 2025-11-05

**Soundness:** 2
**Presentation:** 3
**Contribution:** 3
**Rating:** 4
**Confidence:** 3

**Summary:**

This paper proposes a novel framework for diverse text generation based on "soft prompts." The core technique involves inserting trainable embeddings into the hidden states of input tokens to guide the language model. This approach aims to enhance the diversity of generated outputs while preserving their quality. The method's effectiveness was demonstrated on three distinct tasks: combinatorial optimization, question generation, and molecule generation. Experimental results show that the soft-prompt framework outperforms standard decoding baselines, such as temperature sampling and nucleus sampling, in achieving a better balance between output diversity and quality.

**Strengths:**

1. Novel Method: Introduces a lightweight and effective method that uses inference-time optimization of soft prompts to explicitly balance generation diversity and quality.

2. Strong Empirical Results: Demonstrates superior performance and generalizability by outperforming strong baselines on the quality-diversity trade-off across three highly distinct tasks.

3. Valuable Contributions: Contributes a new benchmark dataset and provides insightful analysis that deepens the community's understanding of controlled diverse generation.

**Weaknesses:**

1. Lack of Ablation Studies: The paper does not sufficiently justify key design choices. A thorough ablation study is needed to validate the contribution of individual components, such as the d_ctrl and w_c terms in the loss function, and to compare the chosen soft-prompt insertion position against alternatives (e.g., prepending to the entire input).

2. Limited Generalizability: The approach appears to require training new, task-specific soft prompts for each distinct task. This limits its generalizability and raises questions about its utility as a universal diversification framework.

3. Unclear Computational Overhead: Given the need for per-task optimization (Algorithm 1, lines 4-20), the paper fails to report the actual wall-clock time required. This omission makes it difficult to assess the practical time cost and evaluate the claim that the framework is "lightweight," especially when adapting to new tasks.

4. Imprecise Analysis of Results: The analysis of the quality-diversity trade-off could be more rigorous. Vague claims like the one on lines 367-368 ("3-5% higher UniEval scores...") should be clarified. The core contribution should be framed as a Pareto improvement—achieving higher diversity for a given level of quality, or vice versa—rather than suggesting a simultaneous improvement in all cases.

**Questions:**

As mentioned in the weakness section.

---

> ### Author Response · Authors · 2025-12-03
> **Official Response to Reviewer UaGR**
>
> Hi Reviewer UaGR,
>
> Thank you so much for your insightful comments! Please find our response to your raised concerns below:
>
> > **Weakness 1: Lack of ablations in hyperparameters in loss function, insertion place of soft prompt**.
> >
>
> $d_\text{ctrl}$ and $w_c$ are empirically chosen hyperparameters, selected through standard tuning to get stable training and a reasonable quality–diversity trade-off. We did not find the method to be very sensitive to their exact values, and changing them mainly scales the same loss terms rather than changing the performance of the method.
>
> As for the insertion point for the soft prompt, we initially experimented with the common approach of prepending soft prompts to the entire input. However, we ultimately chose to insert soft prompts between the context prompt and the generation prompt for both **descent task alignment and efficiency:**
>
> 1. Maintaining task alignment:
>
>     If the input follows the format (soft prompt + context prompt + generation prompt), the soft prompt might influence the interpretation of the context prompt itself from the language model. This may result in generations that deviate from the intended task (e.g., misinterpreting instructions). By placing the soft prompt after the context prompt, we preserve the meaning and role of the original task instructions.
>
> 2. Efficiency via key-value (KV) cache reuse:
>
>     Following 1), since the context prompt remains fixed during soft prompt optimization, as we placed the soft prompt after it, it allows us to reuse the KV cache for the context prompt throughout training. This leads to significant resource savings (with Question Generation Task as an example): Placing the soft prompt at the beginning results in approximately 1.67× speedup in terms of training time. It also leads to ~22% reduction in peak GPU memory usage.
>
>     Note that the magnitude of both time and memory savings **scales with the length of the context prompt**, making this placement particularly beneficial in tasks with long instruction segments. Due to time constraints and the high computational cost of placing soft prompts before the context, we have not yet tested this setup. However, we will include the results in the future appendix.
>
>
> > **Weakness 2:** **Limited Generalizability**
> >
>
> While our diversification framework is general, task-agnostic, and easily transferable across domains, the **soft prompts themselves are intentionally task-specific**, because what counts as “good” diversity is constrained by task structure and can even shift over time. That said, we still believe that **families of soft prompts** can be learned and reused across a broad range of tasks, as suggested by the observed transferability of soft prompts within related task families.
>
> As stated in our discussion section in the manuscript, we believe that task-specific soft prompts are necessary to achieve effective and meaningful diversification. Each task benefits from a tailored diversification strategy rather than relying on a generic one-size-fits-all soft prompt. We also empirically observe that soft prompts can be transferable between similar tasks, such as question generation tasks with different answerability conditions, while different tasks prefer different soft-prompt steering directions to achieve high-quality diversity.
>
> Another perspective showing that a universal diversification strategy is not ideal comes from RL post-training application scenarios, inspired by Reviewer W9rH. In that setting, diversity is not only desired but crucial: soft prompts can be used to encourage exploration during RL-based LM post-training. As training progresses, the “right” kind of diversity changes. This time-varying desire for diversity further shows that steering soft prompts should be adaptable or even learnable. Our framework is compatible with such adaptive strategies, but a detailed treatment of this RL setting is beyond the scope of the current paper and is a promising direction for future work.
>
> However, at the algorithm and framework level, our method does achieve generalizability. To transfer across tasks, we do not need to change model weights, design task-specific rewards, or require task-specific training data. The framework itself is task-agnostic and easily adaptable.

---

> > ### Author Response · Authors · 2025-12-03
> > **Official Response to Reviewer UaGR - continued**
> >
> > Nonetheless, we still believe that families of sets of diversity prompts are plausible. Each set is shared within related task clusters. But to do so, we would need to study what signals in text (or in the model’s latent space) correspond to more universal notions of diversification, and how to cluster tasks based on their diversification objectives and constraints. However, such systematic investigation of such universal signals and task clusters is beyond the scope of this paper. In our work, we introduce a general diversification method that is easily adaptable to downstream tasks without requiring training data. Additionally, we provide a novel knapsack-style combinatorial task dataset that asks an LM to select diverse sets of objects whose values sum to a target value. We show that this task is particularly challenging for recent LMs and that our diversification method substantially improves performance.
> >
> > > **Weakness 3: Unclear Computational Overhead**
> > >
> >
> > Our method adds a soft-prompt training phase compared to purely sampling-based baselines, but this overhead is modest, scales linearly with the number of soft prompts, and is paid only once per task. To make this precise, we report amortized per-instance time under the same number of generations as in Table 2 in the manuscript. For the sampling baselines, this is simply the time to generate all samples for one instance; for our method, it is the training time for the soft prompts plus the subsequent inference time. On a single NVIDIA A100 GPU, the per-instance time for pure sampling vs. our method is approximately 8.4 s vs. 87 s on the combinatorial task, 2.5 s vs. 13.0 s on question generation, and 6.1 s vs. 38.6 s on molecule generation.
> >
> > > **Weakness 4: Imprecise Analysis of Results.**
> > >
> >
> > Due to page limits, we cannot report every specific numerical increment in the main text, though we will still make an effort to be more specific about representative gains. Moreover, we already describe our results in terms of Pareto improvements on the quality–diversity trade-off when reporting combinatorial task results, but we will further clarify and emphasize this framing over the whole result section.

---

### Author Response · Authors · 2025-12-03
**Summary for Area Chairs**

We understand this year’s unpredicted challenges and sincerely appreciate the AC’s additional efforts. Below we briefly summarize the highlights from reviews.

## **Reviews in Strengths**

We introduce a parameter-efficient, task-agnostic soft prompt tuning framework that enables LMs to generate diverse outputs in multi-selection, open-ended scenarios, achieving **Pareto improvements** in the quality–diversity trade-off (recognized by Reviewer Pixg) and **promising results** across benchmarks (recognized by Reviewer W9rH and tcyn). The framework is also **parameter efficient** (recognized by Reviewer Pixg) and is **easily applicable to downstream tasks** without requiring task-specific training data (recognized by Reviewer vCio and W9rH). Additionally, we provide a **new knapsack-style combinatorial benchmark** that offers a challenging and realistic testbed for diversification methods (recognized by Reviewer UaGR).

## **Reviews in Concerns/ Questions**

Across the five reviewers, 15 concerns/questions were raised. 3 of them include additional important experiments, which we have conducted and reported in the rebuttal. Below we summarize common concerns as well as additional experiments we added.

### Clarifications to Major Concerns

- **Generalizability:** Reviewer UGaR, Pixg, and W9rH all raise concerns about the generalizability of soft prompts, in particular, whether there could be a set of prompts trained once and then used across a wide range of tasks for diversification. **We note that the soft prompt training framework in our work is generalizable and does not require external training data.** However, we do not expect a one-size-fits-all diversification soft prompt can be learned because diversification objectives vary across tasks. Instead, we believe a family of sets soft prompts shared within related task clusters might be possible, but this would require systematic investigation into what signals in text or hidden states correspond to more universal diversification behaviors and how to cluster tasks by their diversification goals, which goes beyond the scope of this submission.
- **Theoretical foundation between hidden states and text diversity**: Reviewer vCio and Pixg believe there should be a more solid theoretical proof between the diversity in hidden and diverse in output text semantics. We clarify that we assumed that changes in hidden states transfer to changes in token distributions, not that diverse hidden states are guaranteed to yield semantically diverse outputs. The transferability of useful diversity is demonstrated empirically by consistent gains on task-specific diversity metrics across our three tasks. Moreover, we did not attempt to formally prove a direct link between representational diversity and semantic diversity because (1) any such proof would be heavily architecture-dependent, and (2) diverse generation tasks involve richer forms of diversity than semantics alone.

### Additional Experiments

- **Soft prompts could improve sample efficiency during RL post-training.** We followed Reviewer W9rH’s suggestion and used soft-prompt–guided exploration when post-training Qwen2-0.5B-Instruct with GRPO on the combinatorial task. With soft prompts guided exploration, the model reaches a best@8 score of 0.86 by step 100, whereas the baseline without soft prompts only reaches a similar level around step 250. This shows the potential of soft prompts in improving exploration efficiency in RL-based LM training.
- **Additional baselines: long-context sequential generation and diverse-prefix methods.** In response to Reviewers W9rH and Pixg, we implemented two more baselines: A long-context sequential baseline, where the model sees its previous outputs in the context and a diverse natural-language prefix baseline, where each sample is generated under a different hand-crafted “diversifying” prefix tailored to the task. The long-context baseline is competitive but potentially incurs higher wall-clock time, and the diverse-prefix baseline increases diversity as expected but does so at a substantial cost in quality.

---

### Meta-Review · Area_Chair_Wo9D · 2026-01-08

**Summary:**

The paper introduces a lightweight, task-agnostic framework designed to solve the "quality-diversity trade-off" in Large Language Model (LLM) generation. While standard methods (like temperature sampling) often sacrifice quality for diversity, and Reinforcement Learning (RL) is computationally expensive, this paper proposes Soft Prompt Tuning.

I agree with reviewers regarding the limited novelty.

**Reviewer Concerns:**

Reviewers (UaGR, W9rH) noted that because soft prompts must be trained for each specific task, the method might not be a "universal" solution. They questioned if a set of "general diversity prompts" could exist.

Reviewers (UaGR, W9rH) were skeptical of the "lightweight" claim, asking for exact wall-clock times for the training phase compared to simple sampling.

Reviewer vCio pointed out a lack of theoretical proof that forcing diversity in hidden representation space guaranteed diversity in semantic token space, especially given the non-linear nature of the Softmax layer.
It was unclear to reviewers exactly how these diverse prompts steer the autoregressive trajectory without breaking the model’s coherence.

Reviewer tcyn argued that soft prompt tuning is a well-established field (citing Lester et al., 2021) and felt the technical delta of this paper was insufficient for a top-tier venue.

Reviewer W9rH suggested comparing the method against sequential generation (where the LLM sees its previous outputs in the context window to avoid repeating itself), which requires no training at all.

Reviewer UaGR noted a lack of testing on different insertion points for the prompts (e.g., prepending to the very beginning vs. the middle) and the impact of specific loss function hyperparameters.

**Reviewer Scores:**

I think most of the reviewers will maintain their original score even if they had been able to participate fully in the discussion.

---

### Decision · Program_Chairs · 2026-01-26

Reject